# Analyzing and Improving Optimal-Transport-based Adversarial Networks

**Jaemoo Choi**[*]
Seoul National University
toony42@snu.ac.kr

**Jaewoong Choi** [*]
Korea Institute for Advanced Study
chwj1475@kias.re.kr

**Myungjoo Kang**
Seoul National University
mkang@snu.ac.kr

## Abstract

Optimal Transport (OT) problem aims to find a transport plan that bridges two distributions while minimizing a given cost function. OT theory has been widely utilized in generative modeling. In the beginning, OT distance has been used as a measure for assessing the distance between data and generated distributions. Recently, OT transport map between data and prior distributions has been utilized as a generative model. These OT-based generative models share a similar adversarial training objective. In this paper, we begin by unifying these OT-based adversarial methods within a single framework. Then, we elucidate the role of each component in training dynamics through a comprehensive analysis of this unified framework. Moreover, we suggest a simple but novel method that improves the previously best-performing OT-based model. Intuitively, our approach conducts a gradual refinement of the generated distribution, progressively aligning it with the data distribution. Our approach achieves a FID score of 2.51 on CIFAR-10 and 5.99 on CelebA-HQ-256, outperforming unified OT-based adversarial approaches.

## 1 Introduction

Optimal Transport (OT) theory addresses the most cost-efficient way to transport one probability distribution to another (Villani et al., 2009; Peyré et al., 2017). OT theory has been widely exploited in various machine learning applications, such as generative modeling (Arjovsky et al., 2017; Rout et al., 2022), domain adaptation (Guan et al., 2021; Shen et al., 2018), unpaired image-to-image translation (Korotin et al., 2023; Xie et al., 2019), point cloud approximation (Mérigot et al., 2021), and data augmentation (Alvarez-Melis & Fusi, 2020; Flamary et al., 2016). In this work, we focus on OT-based generative modeling. During its early stages, WGAN (Arjovsky et al., 2017) and its variants (Gulrajani et al., 2017; Petzka et al., 2018; Liu et al., 2019; Miyato et al., 2018) introduced OT theory to define loss functions in GANs (Goodfellow et al., 2020) (*OT Loss*). More precisely, OT-based Wasserstein distance is introduced for measuring a distance between data and generated distributions. These approaches have shown relative stability and improved performance compared to the vanilla GAN (Gulrajani et al., 2017). However, these models still face challenges, such as an unstable training process and limited expressivity (Sanjabi et al., 2018; Mescheder et al., 2018).

Recently, an alternative approach has been introduced in OT-based generative modeling. These works consider OT problems between noise prior and data distributions, aiming to learn the transport map between them (An et al., 2020a;b; Fan et al., 2022) (*OT Map*). This transport map serves as a generative model because it moves a noise sample into a data sample. In this context, two noteworthy methods have been proposed: OTM (Rout et al., 2022) and UOTM (Choi et al., 2023a). Interestingly, these two algorithms present similar adversarial training algorithms as previous OT Loss approaches, like WGAN, but with additional cost function and composition with convex functions (Eq. 5 and 8). These models, especially UOTM, demonstrated promising outcomes, particularly in terms of stability in convergence and performance. Nevertheless, despite the success of OT Map approaches, there is a lack of understanding about why they achieve such high performance and what their limitations are.

(i) In this paper, we propose a unified framework that integrates previous OT Loss and OT Map approaches. Since both of these approaches utilize GAN-like adversarial training, we collectively

---

[*]Equal contribution. Correspondence to: Myungjoo Kang <mkang@snu.ac.kr>.

refer to them as **OT-based GANs**. (ii) Utilizing this framework, we conduct a comprehensive analysis of previous OT-based GANs for an in-depth analysis of each constituent factor of OT Map. Our analysis reveals that the cost function mitigates the mode collapse problem, and the incorporation of a strictly convex function into discriminator loss is beneficial for the stability of the algorithm. (iii) Moreover, we propose a straightforward but novel method for improving the previous best-performing OT-based GANs, i.e., UOTM. Our method involves a gradual up-weighting of divergence terms in the Unbalanced Optimal Transport problem. In this respect, we refer to our model as *UOTM with Scheduled Divergence (UOTM-SD)*. This gradual up-weighting of divergence terms in UOTM-SD leads to the convergence of the optimal transport plan from the UOT problem toward that of the OT problem. Our UOTM-SD outperforms UOTM and significantly improves the sensitivity of UOTM to the cost-intensity hyperparameter. Our contributions can be summarized as follows:

- We introduce an integrated framework that encompasses previous OT-based GANs.
- We present a comparative analysis of these OT-based GANs to elucidate the role of each component.
- We propose a simple and well-motivated modification to UOTM that improves both generation results and $\tau$-robustness of UOTM for the cost function $c(x, y) = \tau \|x - y\|_2^2$.

**Notations** Let $\mathcal{X}$, $\mathcal{Y}$ be compact Polish spaces and $\mu$ and $\nu$ be probability distributions on $\mathcal{X}$ and $\mathcal{Y}$, respectively. Assume that these probability spaces satisfy some regularity conditions described in Appendix A. We denote the prior (source) distribution as $\mu$ and data (target) distributions as $\nu$. $\mathcal{M}_+(\mathcal{X} \times \mathcal{Y})$ represents the set of positive measures on $\mathcal{X} \times \mathcal{Y}$. For $\pi \in \mathcal{M}_+(\mathcal{X} \times \mathcal{Y})$, we denote the marginals with respect to $\mathcal{X}$ and $\mathcal{Y}$ as $\pi_0$ and $\pi_1$. $\Pi(\mu, \nu)$ denote the set of joint probability distributions on $\mathcal{X} \times \mathcal{Y}$ whose marginals are $\mu$ and $\nu$, respectively. For a measurable map $T : \mathcal{X} \to \mathcal{Y}$, $T_{\#}\mu$ denotes the associated pushforward distribution of $\mu$. $c(x, y)$ refers to the transport cost function defined on $\mathcal{X} \times \mathcal{Y}$. In this paper, we assume $\mathcal{X}, \mathcal{Y} \subset \mathbb{R}^d$ and the quadratic cost $c(x, y) = \tau \|x - y\|_2^2$, where $\tau$ is a given positive constant. For a detailed explanation of assumptions, see Appendix A.

## 2 BACKGROUND AND RELATED WORKS

In this section, we introduce several OT problems and their equivalent forms. Then, we provide an overview of various OT-based GANs, which will be the subject of our analysis.

**Kantorovich OT** Kantorovich (1948) formulated the OT problem through the cost-minimizing coupling $\pi \in \Pi(\mu, \nu)$ between the source distribution $\mu$ and the target distribution $\nu$ as follows:

$$C(\mu, \nu) := \inf_{\pi \in \Pi(\mu, \nu)} \left[ \iint_{\mathcal{X} \times \mathcal{Y}} c(x, y) \, \mathrm{d}\pi(x, y) \right], \tag{1}$$

Under mild assumptions, this Kantorovich problem can be reformulated into several equivalent forms, such as the *dual* (Eq. 2) and *semi-dual* (Eq. 3) formulation (Villani et al., 2009):

$$C(\mu, \nu) = \sup_{u(x) + v(y) \le c(x,y)} \left[ \int_{\mathcal{X}} u(x) \, \mathrm{d}\mu(x) + \int_{\mathcal{Y}} v(y) \, \mathrm{d}\nu(y) \right], \tag{2}$$

$$= \sup_{v \in L^1(\nu)} \left[ \int_{\mathcal{X}} v^c(x) \, \mathrm{d}\mu(x) + \int_{\mathcal{Y}} v(y) \, \mathrm{d}\nu(y) \right], \tag{3}$$

where $u$ and $v$ are Lebesgue integrable with respect to measure $\mu$ and $\nu$, i.e., $u \in L^1(\mu)$ and $v \in L^1(\nu)$, and the $c$-transform $v^c(x) := \inf_{y \in \mathcal{Y}} (c(x, y) - v(y))$. For a particular case where the cost $c(\cdot, \cdot)$ is a distance function, i.e., the Wasserstein-1 distance, then $u = -v$ and $u$ is 1-Lipschitz (Villani et al., 2009). In such case, we call Eq. 2 a *Kantorovich-Rubinstein duality*.

**OT Loss in GANs** WGAN (Arjovsky et al., 2017) introduced the Wasserstein-1 distance to define a loss function in GAN. This Wasserstein distance serves as a distance measure between generated distribution and data distribution. From Kantorovich-Rubinstein duality, the optimization problem for WGAN is given as follows:

$$\mathcal{L}_{v_\phi, T_\theta} = \sup_{\|v_\phi\|_L \le 1} \inf_{T_\theta} \left[ -\int_{\mathcal{X}} v_\phi (T_\theta(x)) \, \mathrm{d}\mu(x) + \int_{\mathcal{Y}} v_\phi(y) \, \mathrm{d}\nu(y) \right], \tag{4}$$

where the potential (critic) $v_\phi$ is 1-Lipschitz, i.e., $\|v_\phi\|_L \leq 1$, and $T_\theta$ is a generator. WGAN-GP (Gulrajani et al., 2017) suggested a gradient penalty regularizer to enhance the stability of WGAN training. For optimal coupling $\pi^*$, the optimal potential $v_\phi^*$ satisfies $\|\nabla v_\phi(\hat{y})\|_2 = 1$ $\pi^*$-almost surely, where $\hat{y} = tT_\theta(x) + (1-t)y$ for some $0 \leq t \leq 1$ with $(T_\theta(x), y) \sim \pi^*$. WGAN-GP exploits this optimality condition by introducing $\mathcal{R}(x,y) = (\|\nabla v_\phi(\hat{y})\|_2 - 1)^2$ as the penalty term.

**OT Map as Generative model** Parallel to OT Loss approaches, there has been a surge of research on directly modeling the optimal transport map between the input prior distribution and the real data distribution (Rout et al., 2022; An et al., 2020a;b; Makkuva et al., 2020; Yang & Uhler, 2019; Choi et al., 2023a). In this case, the optimal transport map serves as the generator itself. In particular, Rout et al. (2022) and Fan et al. (2022) leverage the semi-dual formulation (Eq. 3) of the Kantorovich problem for generative modeling. Specifically, these models parametrize $v = v_\phi$ in Eq. 3 and represent its $c$-transform $v^c$ through the transport map[1] $T_\theta : \mathcal{X} \to \mathcal{Y}$, $x \mapsto \arg\inf_{y \in \mathcal{Y}} [c(x,y) - v(y)]$. Then, we obtain the following optimization problem:

$$\mathcal{L}_{v_\phi, T_\theta} = \sup_{v_\phi} \left[ \int_\mathcal{X} \inf_{T_\theta} \left[ c\left(x, T_\theta(x)\right) - v_\phi\left(T_\theta(x)\right) \right] \mathrm{d}\mu(x) + \int_\mathcal{Y} v_\phi(y)\, \mathrm{d}\nu(y) \right]. \quad (5)$$

Intuitively, $T_\theta$ and $v_\phi$ serve as the generator and the discriminator of a GAN. For convenience, we denote the optimization problem of Eq. 5 as an OT-based generative model (OTM) (Fan et al., 2022). *Note that, if we set $c = 0$ and introduce a 1-Lipschitz constraint on $v_\phi$, this objective has the same form as WGAN (Eq. 4).* In OT map models, the quadratic cost is usually employed.

Recently, Choi et al. (2023a) extended OTM by leveraging the semi-dual form of the Unbalanced Optimal Transport (UOT) problem (Liero et al., 2018). UOT extends the OT problem by relaxing the strict marginal constraints using the Csiszàr divergences $D_{\Psi_i}$ (See the Appendix A for precise definition). Formally, the UOT problem (Eq. 6) and its semi-dual form (Eq. 7) are defined as follows:

$$C_{ub}(\mu, \nu) = \inf_{\pi \in \mathcal{M}_+(\mathcal{X} \times \mathcal{Y})} \left[ \int_{\mathcal{X} \times \mathcal{Y}} c(x,y)\, \mathrm{d}\pi(x,y) + D_{\Psi_1}(\pi_0 | \mu) + D_{\Psi_2}(\pi_1 | \nu) \right], \quad (6)$$

$$= \sup_{v \in \mathcal{C}(\mathcal{Y})} \left[ \int_\mathcal{X} -\Psi_1^*\left( -v^c(x) \right) \mathrm{d}\mu(x) + \int_\mathcal{Y} -\Psi_2^*(-v(y))\, \mathrm{d}\nu(y) \right], \quad (7)$$

where $\mathcal{C}(\mathcal{Y})$ denotes a set of continuous functions over $\mathcal{Y}$. Here, the entropy function $\Psi_i : \mathbb{R} \to [0, \infty]$ is a convex, lower semi-continuous, and non-negative function, and $\Psi_i(x) = \infty$ for $x < 0$. $\Psi_i^*$ denotes its convex conjugate. Note that for non-negative $\Psi_i$, $\Psi_i^*$ **is a non-decreasinig convex function**. For simplicity, we assume $\Psi_i^*(0) = 0, (\Psi_i^*)'(0) = 1$. The reason for this assumption will be clarified in Sec 4. By using the same parametrization as in Eq. 5, we arrive at the following:

$$\mathcal{L}_{v_\phi, T_\theta} = \inf_{v_\phi} \left[ \int_\mathcal{X} \Psi_1^* \left( -\inf_{T_\theta} \left[ c\left(x, T_\theta(x)\right) - v_\phi\left(T_\theta(x)\right) \right] \right) \mathrm{d}\mu(x) + \int_\mathcal{Y} \Psi_2^*\left( -v_\phi(y) \right) \mathrm{d}\nu(y) \right]. \quad (8)$$

We call such an optimization problem a UOT-based generative model (UOTM) (Choi et al., 2023a).

## 3 ANALYZING OT-BASED ADVERSARIAL APPROACHES

In this section, we suggest a unified framework for OT-based GANs (Sec 3.1). Using this unified framework, we compare the dynamics of each algorithm through various experimental results (Sec 3.2). This comparative ablation study delves into the impact of employing strictly convex $g_1, g_2$ within discriminator loss, and the influence of cost function $c(x,y) = \tau \|x - y\|_2^2$. Furthermore, we present an additional explanation for the success of UOTM (Sec 3.2.3).

### 3.1 A UNIFIED FRAMEWORK

We present an integrated framework, Algorithm 1, that includes various OT-based adversarial networks. These models are derived by directly parameterizing the potential and generator, utilizing the

---

[1]Note that this parametrization does not precisely characterize the optimal transport map (Rout et al., 2022). The optimal transport map satisfies this relationship, but not all functions satisfying this condition are transport maps. However, investigating a better parametrization of the optimal transport is beyond the scope of this work.

---

**Algorithm 1** Unified training algorithm

---

**Require:** Functions $g_1, g_2, g_3$. Generator network $T_\theta$ and the discriminator network $v_\phi$. The number of iterations per network $K_v, K_T$. Total iteration number $K$. Regularizer $\mathcal{R}$ with regularization hyperparameter $\lambda$.
1: **for** $k = 0, 1, 2, \ldots, K$ **do**
2:      **for** $k = 1$ to $K_v$ **do**
3:          Sample a batch $X \sim \mu$, $Y \sim \nu$, $z \sim \mathcal{N}(\mathbf{0}, \mathbf{I})$.
4:          $\hat{y} = T_\theta(x, z)$.
5:          $\mathcal{L}_v = \frac{1}{|X|} \sum_{x \in X} g_1 \left(-c(x, \hat{y}) + v_\phi(\hat{y})\right) + \frac{1}{|Y|} \sum_{y \in Y} g_2(-v_\phi(y)) + \lambda \mathcal{R}(y, \hat{y})$.
6:          Update $\phi$ by using the loss $\mathcal{L}_v$.
7:      **end for**
8:      **for** $k = 1$ to $K_T$ **do**
9:          Sample a batch $X \sim \mu$, $z \sim \mathcal{N}(\mathbf{0}, \mathbf{I})$.
10:         $\mathcal{L}_T = \frac{1}{|X|} \sum_{x \in X} g_3((c(x, T_\theta(x, z)) - v_\phi(T_\theta(x, z))))$.
11:         Update $\theta$ by using the loss $\mathcal{L}_T$.
12:      **end for**
13: **end for**

---

dual or semi-dual formulations of OT or UOT problems. Specifically, Algorithm 1 can represent the following models, depending on *the choice of the cost function $c(x, y)$, the convex functions $g_1, g_2, g_3$, and the regularization term $\mathcal{R}$*. (Note that $g_1, g_2$ correspond to $\Psi_{1,2}^*$ in Eq 8.) Here, we denote two convex functions, Identity and Softplus, as $\mathrm{Id}(x) = x$ and $\mathrm{SP}(x) = 2\log(1 + e^x) - 2\log 2$.[2] Also, the Gaussian noise $z$ represents the auxiliary variable and is different from the input prior noise $x \sim \mu$. This auxiliary variable $z$ is introduced to represent the stochastic transport map $T_\theta$ in the OT map models, such as UOTM.

- **WGAN** (Arjovsky et al., 2017) if $c \equiv 0$ and $g_1 = g_2 = g_3 = \mathrm{Id}$.[3]
- **WGAN-GP** (Gulrajani et al., 2017) if $c \equiv 0$, $g_1 = g_2 = g_3 = \mathrm{Id}$, and $\mathcal{R}$ a gradient penalty.
- **OTM** (Rout et al., 2022) if $\tau > 0$ and $g_1 = g_2 = g_3 = \mathrm{Id}$.
- **UOTM** (Choi et al., 2023a) if $\tau > 0$, $g_1 = g_2 = \mathrm{SP}$ and $g_3 = \mathrm{Id}$.
- **UOTM w/o cost** if $\tau = 0$, $g_1 = g_2 = \mathrm{SP}$ and $g_3 = \mathrm{Id}$.

In this work, we conduct a comprehensive comparative analysis of OT-based GANs: ***WGAN, OTM, UOTM w/o cost, and UOTM*** (Table 1). This comparative analysis serves as an ablation study of two building blocks of OT-based GANs. Thus, we focus on investigating the influence of cost $c(\cdot, \cdot)$ and $g_1, g_2$.

Table 1: **Unified Framework for OT-based GANs** ($g_3 = \mathrm{Id}$).

| | $g_1 = g_2 = \mathrm{Id}$ | $g_1 = g_2 = \mathrm{SP}$ |
|---|---|---|
| $\tau = 0$ | WGAN | UOTM w/o cost |
| $\tau > 0$ | OTM | UOTM |

## 3.2 COMPARATIVE ANALYSIS OF OT-BASED GANs

In this section, we present qualitative and quantitative generation results of OT-based GANs on both toy and CIFAR-10 (Krizhevsky et al., 2009) datasets. We particularly discuss how the algorithms differ with respect to functions $g_1 \& g_2$, and the cost $c(\cdot, \cdot)$. This analysis is conducted in terms of the well-known challenges associated with adversarial training procedures, namely, *Unstable training and Mode collapse/mixture*. (See Appendix C for the introduction of these challenges.) Moreover, we provide an in-depth analysis of the underlying reasons behind these observed phenomena.

**Experimental Settings** For visual analysis of the training dynamics, we evaluated these models on 2D multivariate Gaussian distribution, where the source distribution $\mu$ is a standard Gaussian. The network architecture is fixed for a fair comparison. Note that we impose $R_1$ regularization (Roth et al., 2017) to WGAN and OTM because they diverge without any regularizations. Moreover, to investigate the scalability of the algorithms, we assessed these models on CIFAR-10 with various network architectures and hyperparameters. See Appendix B for detailed experiment settings.

---

[2]The softplus function is scaled and translated to satisfy $\mathrm{SP}(0) = 0$ and $\mathrm{SP}'(0) = 1$.
[3]For the vanilla WGAN, we employed a weight clipping strategy following Arjovsky et al. (2017).

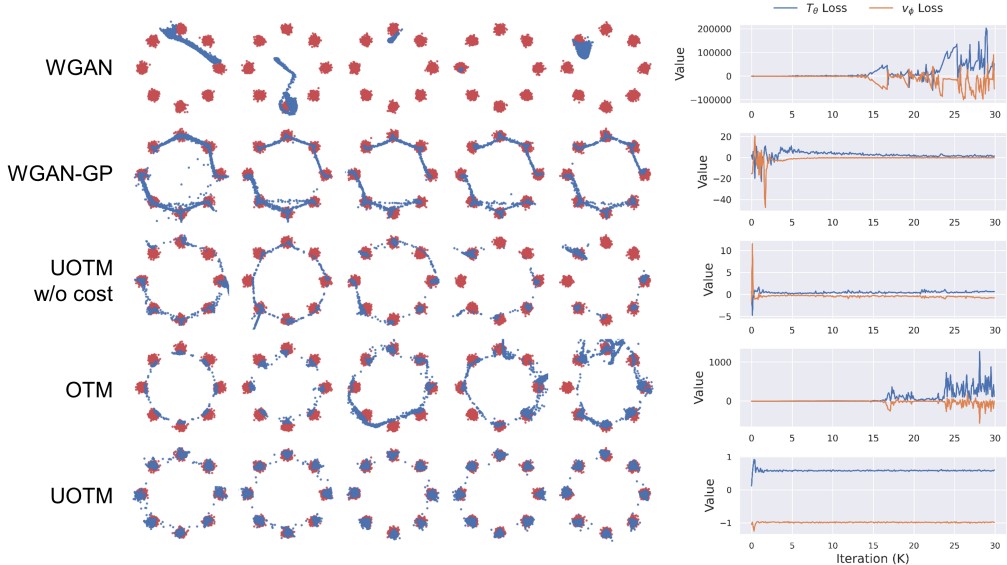

Figure 1: **Comparison of Training Dynamics between OT-based GANs. Left**: Visualization of generated samples (blue) and data samples (red) for every 6K iterations. **Right**: Training loss of the generator ($T_\theta$ Loss) and discriminator ($v_\phi$ Loss) for each algorithm.

Table 2: **Quantitative Evaluation of OT-based GANs** on CIFAR-10.

| Model | Metric | | |
|---|---|---|---|
| | FID ($\downarrow$) | Precision ($\uparrow$) | Recall ($\uparrow$) |
| WGAN | 48.8 | 0.45 | 0.02 |
| WGAN-GP | 4.5 | 0.71 | 0.55 |
| OTM | 4.3 | 0.71 | 0.49 |
| UOTM w/o cost | 19.7 | **0.80** | 0.13 |
| UOTM (SP) | **2.7** | 0.78 | **0.62** |
| UOTM (KL) | 2.9 | - | - |

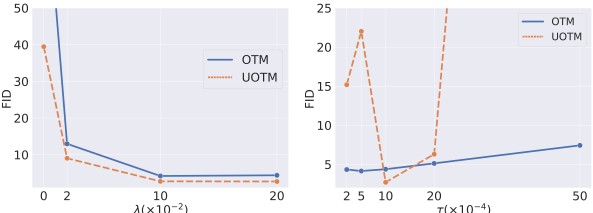

Figure 2: **Ablation Study on Regularizer Intensity $\lambda$.** Figure 3: **Ablation Study on Cost Intensity $\tau$.**

### 3.2.1 EFFECT OF STRICTLY CONVEX $g_1$ AND $g_2$

**Experimental Results** Fig 1 illustrates how each model evolves during training for each 6K iterations. To investigate the effect of $g_1$ and $g_2$, we compare the models with $g_1 = g_2 = \mathrm{Id}$ (WGAN, OTM) and $g_1 = g_2 = \mathrm{Sp}$ (UOTM, UOTM w/o cost). When $g_1 = g_2 = \mathrm{Id}$, WGAN and OTM initially appear to converge in the early stages of training. However, as training progresses, the loss highly fluctuates, leading to divergent results. Interestingly, adding a gradient penalty regularizer to WGAN (WGAN-GP) is helpful in addressing this loss fluctuation. Conversely, when $g_1 = g_2 = \mathrm{Sp}$, UOTM and UOTM w/o cost consistently perform well, with the loss steadily converging during training. From these observations, we interpret that **setting $g_1, g_2$ to Sp functions, which are strictly convex, contribute to the stable convergence of OT-based GANs.**

Moreover, Tab 2 presents CIFAR-10 generation results of OT-based GANs with NCSN++ (Song et al., 2021b) backbone architecture (See the Appendix D.2 for DCGAN (Radford et al., 2015) backbone results). Here, we additionally compared UOTM (KL) following Choi et al. (2023a). UOTM (KL) serves as another example of strictly convex $g_1, g_2$, where $g_1 = g_2 = e^x - 1$. [4] As in the toy dataset, UOTM w/o cost and UOTM achieve better FID scores than their algorithmic counterparts with respect to $g_1, g_2$, i.e., WGAN and OTM, respectively. The precision and recall metric (Kynkäänniemi et al., 2019) results will be examined regarding the cost function in Sec 3.2.2. The additional stability of UOTM can be observed in an ablation study on the regularizer intensity $\lambda$ (Fig 2). UOTM model provides more robust FID results compared to OTM.

---

[4] Here, UOTM (SP) outperformed the original UOTM (KL). Since UOTM defines $g_1, g_2$ as any non-decreasing and convex functions, we adopt UOTM (SP) as the default UOTM model throughout this paper.

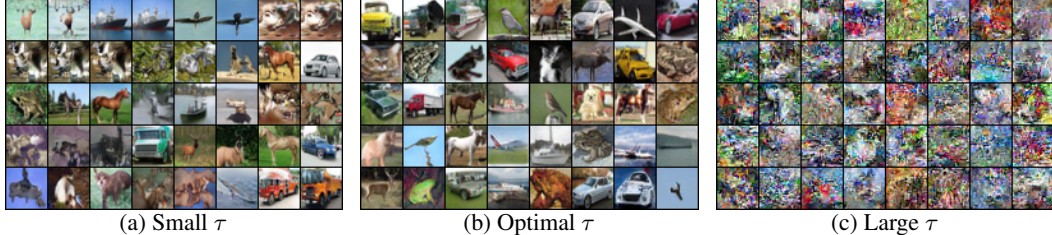

| (a) Small $\tau$ | (b) Optimal $\tau$ | (c) Large $\tau$ |

Figure 4: **Qualitative Comparison of Generated Samples from UOTM. Left**: When $\tau$ is too small ($\tau = 0.0002$). **Middle**: When $\tau$ is optimal ($\tau = 0.001$). **Right**: When $\tau$ is too large ($\tau = 0.005$). On Left, we reordered randomly generated samples to gather similar-looking samples. When $\tau$ is large, the samples appear noisy, and when $\tau$ is small, the generated samples show a mode collapse problem.

**Effect of $g_1$ and $g_2$ in Optimization** We observed that the introduction of strictly convex functions, such as SP or $g(x) = e^x - 1$, into $g_1, g_2$ contributes to more stable training of OT-based GANs. We explain this enhanced stability in terms of **the adaptive optimization of the potential network** $v_\phi$. From the potential loss function $\mathcal{L}_v$ in line 5 of Algorithm 1, we can express the gradient descent update for the potential function $v_\phi$ with a learning rate $\gamma$ as follows:

$$\phi - \gamma \nabla_\phi \mathcal{L}_{v_\phi} = \phi - \frac{\gamma}{|X|} \sum_{x \in X} \underbrace{g_1'(-\hat{l}(x))}_{=: \hat{w}(x)} \nabla_\phi v_\phi(\hat{y}) + \frac{\gamma}{|Y|} \sum_{y \in Y} \underbrace{g_2'(-v_\phi(y))}_{=: w(y)} \nabla_\phi v_\phi(y), \quad (9)$$

where $\hat{l}(x) = c(x, \hat{y}) - v_\phi(\hat{y})$. Note that the generator loss $\mathcal{L}_T = \frac{1}{|X|} \sum_{x \in X} \hat{l}(x)$, since we assume $g_3 = \text{Id}$. Here, $w$ and $\hat{w}$ in Eq. 9 serve as sample-wise weights for the potential gradient $\nabla_\phi v_\phi$.

*We interpret the role of $g_1, g_2$ as mediating the balance between $T$ and $v$.* Suppose the generator dominates the potential for certain $x$, i.e., $\hat{l}(x)$ is small. In this case, because $g_1'$ is a strictly increasing function, the weight $\hat{w}(x)$ becomes large for this sample $x$, counterbalancing the dominant generator. Similarly, consider the weight of the true data sample $w(y)$. Note that the goal of potential is to assign a high value to real data $y$ and a low value to generated samples $\hat{y}$. Assume that the potential is not good at discriminating certain $y$, which means that $v(y)$ is small. Then, the weight $w(y)$ becomes large for this sample $y$ as above. We hypothesize that this failure-aware adaptive optimization of the potential $v_\phi$ stabilizes the training procedure, regardless of the regularizer.

### 3.2.2 EFFECT OF COST FUNCTION

**Experimental Results** To examine the effect of the cost function $c(x, y) = \tau \|x - y\|_2^2$, we compare the models with $\tau = 0$ (WGAN, UOTM w/o cost) and $\tau > 0$ (OTM, UOTM) in Fig 1. When $\tau = 0$, both WGAN and UOTM w/o cost exhibit a mode collapse problem. These models fail to fit all modes of the data distribution. On the other hand, WGAN-GP shows a mode mixture problem. WGAN-GP generates inaccurate samples that lie between the modes of data distribution. In contrast, when $\tau > 0$, both OTM and UOTM avoid model collapse and mixture problems. In the initial stages of training, OTM succeeds in capturing all modes of data distribution, until training instability occurs due to loss fluctuation. UOTM achieves the best distribution fitting by exploiting the stability of $g_1, g_2$ as well. Moreover, Table 2 provides a quantitative assessment of the mode collapse problem on CIFAR-10. The results are consistent with our analysis on the Toy datasets (Fig 1). The recall metric assesses the mode coverage for each model. In this regard, the introduction of the cost function improves the recall metric for each model: from WGAN (0.02) to OTM (0.49) and from UOTM w/o cost (0.13) to UOTM (0.62). The precision metric evaluates the faithfulness of generated images for each model. UOTM w/o cost achieves the best precision score, but the recall metric is significantly lower than UOTM. This result shows that UOTM w/o cost exhibited the mode collapse problem. From these results, we interpret that **the cost function $c(x, y)$ plays a crucial role in preventing mode collapse by guiding the generator towards cost-minimizing pairs**.

Furthermore, we analyze the *influence of the cost function intensity $\tau$* by performing an ablation study on $\tau$ on CIFAR-10 (Fig 3). Interestingly, the results are quite different between OTM and UOTM. When we compare the best-performing $\tau$, UOTM achieves much better FID scores than OTM ($\tau = 10 \times 10^{-4}$). However, when $\tau$ is excessively small or large, the performance of UOTM deteriorates severely. On the contrary, OTM maintains relatively stable results across a wide range of $\tau$. The deterioration of UOTM can be understood intuitively by examining the generated results in

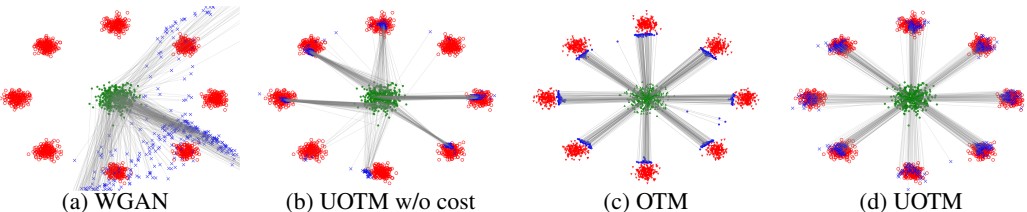

Figure 5: **Visualization of Generator** $T_\theta$**.** The gray lines illustrate the generated pairs, i.e., the connecting lines between $x$ (green) and $T_\theta(x)$ (blue). The red dots represent the training data samples.

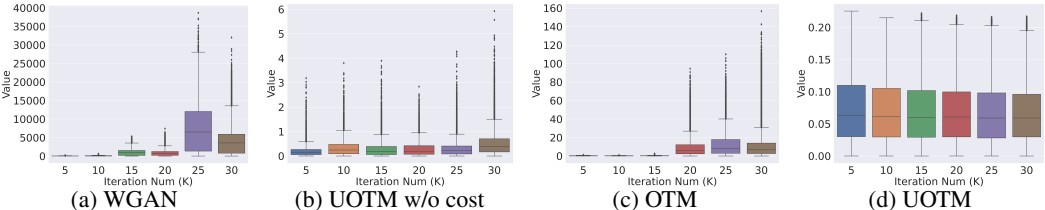

Figure 6: **Distribution of the absolute value of Average Rate of Change (ARC) of potential** $\frac{|v_\phi(y)-v_\phi(x)|}{\|y-x\|}$ for every 5K iterations. Due to the equi-Lipschitz property, |ARC| of UOTM potential is stable during training. This stability contributes to the stable training of UOTM.

Fig 4. When $\tau$ is too large, UOTM tends to produce noise-like samples because the cost function dominates the other divergence terms $D_{\Psi_i}$ within the UOT objective (Eq. 6). When $\tau$ is too small, UOTM shows a mode collapse problem because the negligible cost function fails to prevent the mode collapse. Conversely, as inferred from OT objective (Eq. 1), the optimal pair $(v^\star, T^\star)$ of OTM remains consistent regardless of variations in $\tau$. Hence, OTM presents relatively consistent performance across changes in $\tau$ (Fig 3). In Sec 4, we propose a method that enhances the $\tau$-robustness of UOTM while also improving its best-case performance.

**Effect of Cost in Mode Collapse/Mixture**    In Fig 1, the OT-based GANs with the cost term (OTM, UOTM) exhibited a significantly lower occurrence of mode collapse/mixture, compared to the models without the cost term. This observation proves that the cost function plays a regularization role in OT-based GANs, helping to cover all modes within the data distribution. This cost function encourages the generator $T$ to minimize the quadratic error between input $x$ and output $T(x)$. **In other words, the generator $T$ is indirectly guided to transport each input $x$ to a point, that is within the data distribution support and close to** $x$. Fig 5 visualizes the transported pair $(x, T(x))$ by the gray line that connects $x$ and $T(x)$. Fig 5 demonstrates that this cost term induces OTM and UOTM to spread the generated samples (in an optimal way). (See Appendix D.1 for a comprehensive comparison of generators, including WGAN-GP, UOTM-SD, and GT transport map.)

### 3.2.3    ADDITIONAL ADVANTAGE OF UOTM

**Lipshitz Continuity of UOTM Potential**    We offer an additional explanation for the stable convergence observed in UOTM. In OT-based GANs, we approximate the generator and potential with neural networks. However, since neural networks can only represent continuous functions, it is crucial to verify the regularity of these target functions, such as Lipschitz continuity. If these target functions are not continuous, the neural network approximation may exhibit highly irregular behavior. Theorem 3.1 proves that under minor assumptions on $g_1$ and $g_2$ in UOTM (Choi et al., 2023a), *there exists unique optimal potential $v^\star$ and it satisfies Lipschitz continuity.* (See Appendix A for the proof.)

**Theorem 3.1.** *Let $g_1$ and $g_2$ be real-valued functions that are non-decreasing, bounded below, differentiable, and strictly convex. Assuming the regularity assumptions in Appendix A, there exists a unique Lipschitz continuous optimal potential $v^\star$ for Eq. 7. Moreover, for the semi-dual maximization objective $-\mathcal{L}_v$ (Eq. 7),*

$$\Gamma := \left\{ v \in \mathcal{C}(\mathcal{Y}) : \mathcal{L}_v \leq 0, v^{cc} = v \right\}, \tag{10}$$

*is equi-bounded and equi-Lipschitz.*

Table 3: **Image Generation on CIFAR-10.** † indicates the results conducted by ourselves.

| Class | Model | FID ($\downarrow$) |
|---|---|---|
| **GAN** | SNGAN+DGflow (Ansari et al., 2020) | 9.62 |
| | StyleGAN2 w/o ADA (Karras et al., 2020) | 8.32 |
| | StyleGAN2 w/ ADA (Karras et al., 2020) | **2.92** |
| | DDGAN (Xiao et al., 2021) | 3.75 |
| | RGM (Choi et al., 2023b) | **2.47** |
| **Diffusion** | DDPM (Ho et al., 2020) | 3.21 |
| | Score SDE (VE) (Song et al., 2021b) | 2.20 |
| | Score SDE (VP) (Song et al., 2021b) | 2.41 |
| | DDIM (50 steps) (Song et al., 2021a) | 4.67 |
| | CLD (Dockhorn et al., 2022) | 2.25 |
| | LSGM (Vahdat et al., 2021) | **2.10** |
| **OT-based** | WGAN (Arjovsky et al., 2017) | 55.20 |
| | WGAN-GP(Gulrajani et al., 2017) | 39.40 |
| | OTM † (Rout et al., 2022) | 4.15 |
| | UOTM (Choi et al., 2023a) | 2.97 |
| | UOTM-SD (Cosine)† | 2.57 |
| | UOTM-SD (Linear)† | **2.51** |
| | UOTM-SD (Step)† | 2.78 |

Table 4: **Image Generation on CelebA-HQ.**

| Class | Model | FID ($\downarrow$) |
|---|---|---|
| **OT-based** | OTM† | 13.56 |
| | UOTM (KL) | 6.36 |
| | UOTM (SP)† | 6.31 |
| | **UOTM-SD**† | **5.99** |

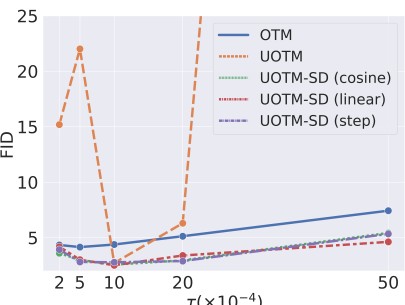

Table 5: **Comparison of $\tau$-robustness.**

Note that the semi-dual objective $\mathcal{L}_v$ can be derived by assuming the optimality of $T_\theta$ for given $v$, i.e., $T_\theta(x) \in \arg\inf_{y \in \mathcal{Y}} [c(x, y) - v(y)]$. Also, Theorem 3.1 shows that the set of valid[5] potential candidates $\Gamma$ is equi-Lipschitz, i.e., there exists a Lipshitz constant $L_\Gamma$ that all $v \in \Gamma$ are $L_\Gamma$-Lipschitz. **This equi-Lipschitz continuity also explains the stable training of UOTM over OTM.** The condition $\mathcal{L}_v \leq 0$ in $\Gamma$ is not a tough condition for the neural network $v_\phi$ to satisfy during training, since $\mathcal{L}_v = 0$ when $v \equiv 0$ [6]. Therefore, during training, the potential network $v_\phi$ would remain within the domain of $L_\Gamma$-Lipshitz functions. In other words, $v_\phi$ would not express any drastic changes for all input $y$. Furthermore, the target of training, $v^\star$, also stays within this set of functions. Hence, we can expect stable convergence of the potential network as training progresses. Note that Theorem 3.1 is fundamentally different from the 1-Lipschitz constraint of WGAN. WGAN involves constrained optimization over a 1-Lipschitz potential. In contrast, Theorem 3.1 states that, under unconstrained optimization, the potential networks $v_\phi$ with only minor conditions satisfy equi-Lipschitzness.

**Experimental Validation** We tested whether this equi-Lipschitz continuity of UOTM potential $v_\phi$ is observed during training in practice. In particular, we randomly choose data $a \in \mathcal{Y}$ and $b \sim \nu$ on 2D experiment and visualize the Average Rate of Change (ARC) of potential $\frac{|v_\phi(b) - v_\phi(a)|}{\|b-a\|}$. Fig 6 shows boxplots of an ARC of ten thousand pairs of $(a, b)$ for every 10K iterations. As shown in Fig 6, only UOTM shows a bounded ARC, and others, especially WGAN and OTM, diverge as the training progresses. This result indirectly shows the potential network in UOTM mostly remains within the equi-Lipschitz set during training. Furthermore, we can conjecture that the highly irregular behavior of potential networks in other models could potentially disrupt stable training processes.

## 4 TOWARDS THE STABLE OT MAP

In this section, we suggest a straightforward yet novel method to enhance the $\tau$-robustness of UOTM, while improving the best-case performance. Intuitively, our idea is to gradually adjust the transport map in the UOT problem towards the transport map in the OT problem. Note that the OT problem of OTM assumes a hard constraint on marginal matching.

**Motivation** The analysis in Sec 3 showed that the semi-dual form of the UOT problem, i.e., UOTM, provides several advantages over other OT-based GANs. However, Fig 4 showed that UOTM is

---

[5]The optimal potential satisfies the $c$-concavity condition $v^{cc} = v$. For the quadratic cost, this is equivalent to the condition that $y \mapsto \frac{\tau}{2}|y|^2 - v(y)$ is convex and lower semi-continuous (Santambrogio, 2015).

[6]In practice, the potential loss $\mathcal{L}_v$ is always $\mathcal{L}_v < 0$ after only 100 iterations.

$\tau$-sensitive. In this respect, Choi et al. (2023a) proved that the upper bound of marginal discrepancies for the optimal $\pi^\star$ in the UOT problem (Eq. 6) is linearly proportional to $\tau$:

$$D_{\Psi_1}(\pi_0^\star|\mu) + D_{\Psi_2}(\pi_1^\star|\nu) \leq \tau \mathcal{W}_2^2(\mu, \nu) \quad \text{for} \quad c(x, y) = \tau\|x - y\|_2^2. \tag{11}$$

When the divergence term is minor (*Large* $\tau$), the cost term prevents the mode collapse problem (Sec 3.2.2), but the model fails to match the target distribution, generating noisy samples (Eq. 11). Conversely, when the divergence term is dominant (*Small* $\tau$), the model should theoretically exhibit improved target distribution matching (Eq. 11, Theorem 4.1). However, the mode collapse problem disturbs the optimization process in practice (Sec 3.2.2). **In this regard, we introduce a method that can leverage the advantages of both regimes: preventing mode collapse with minor divergence and improving distribution matching with dominant divergence.** Intuitively, we start training with a smaller divergence term to mitigate mode collapse. Subsequently, as training progresses, we gradually increase the influence of the divergence term to achieve better data distribution matching.

**Method** Formally, we consider the following $\alpha$-*scaled UOT problem* ($\alpha$-*UOT*) $C_{ub}^\alpha(\mu, \nu)$ for $\alpha \geq 0$ (Eq. D.4). Note that this $\alpha$-UOT problem recovers the OT problem $C(\mu, \nu)$ when $\alpha \to \infty$ if $\mu, \nu$ have equal mass (Fatras et al., 2021).

$$C_{ub}^\alpha(\mu, \nu) = \inf_{\pi^\alpha \in \mathcal{M}_+(\mathcal{X} \times \mathcal{Y})} \left[ \int_{\mathcal{X} \times \mathcal{Y}} c(x, y) \, d\pi^\alpha(x, y) + \alpha D_{\Psi_1}(\pi_0^\alpha|\mu) + \alpha D_{\Psi_2}(\pi_1^\alpha|\nu) \right]. \tag{12}$$

Motivated by this fact, we suggest a monotone-increasing scheduling scheme during training for $\alpha$ to achieve the stable convergence of the UOT transport map $\pi^\alpha$ towards the OT transport map. Because $\alpha D_{\Psi i} = D_{\alpha \Psi i}$ and $(\alpha \Psi_i)^*(x) = \alpha \Psi_i^*(x/\alpha)$, the learning objective of $\alpha$-scaled UOTM are given as follows:

$$\mathcal{L}_{v_\phi, T_\theta}^\alpha = \inf_{v_\phi} \left[ \int_{\mathcal{X}} \alpha \Psi_1^* \left( -\frac{1}{\alpha} \inf_{T_\theta} [c(x, T_\theta(x)) - v(T_\theta(x))] \right) d\mu(x) + \int_{\mathcal{Y}} \alpha \Psi_2^* \left( -\frac{1}{\alpha} v(y) \right) \right]. \tag{13}$$

Note that, given our assumption that $\Psi_i^*$ is $C^1$, $(\alpha \Psi_i)^*$ uniformly converges to Id for every compact domain, since $\Psi_i^*(0) = 0, (\Psi_i^*)'(0) = 1$. Therefore, this $\alpha$-scheduling can be intuitively understood as a gradual process of straightening the strictly convex $\Psi_i^*$ function towards the identity function Id, so that $\mathcal{L}_{v_\phi, T_\theta}^\alpha$ converges to OTM (Tab 1). We refer to this UOTM with $\alpha$-scheduling as **UOTM with Scheduled Divergence (UOTM-SD)**.

**Convergence** Theorem 4.1 proves that the optimal transport plan of the $\alpha$-scaled UOT problem converges to that of the OT problem as $\alpha \to \infty$. However, one limitation of this theorem is that it shows the convergence of transport plan $\pi$, but does not address the convergence of transport map $T$.

**Theorem 4.1.** *Assume the entropy functions $\Psi_1, \Psi_2$ are strictly convex and finite on $(0, \infty)$. Then, the optimal transport plan $\pi^{\alpha,\star}$ of the $\alpha$-scaled UOT problem $C_{ub}^\alpha(\mu, \nu)$ (Eq. 6) weakly converges to the optimal transport plan $\pi^\star$ of the OT problem $C(\mu, \nu)$ (Eq. 1) as $\alpha$ goes to infinity.*

$\alpha$-**schedule Settings** We evaluated three scheduling schemes for $\alpha$. For the schedule parameters $\alpha_{max} \geq \alpha_{min} > 0$, the assessed scheduling schemes are as follows:

- **Cosine** : Apply Cosine scheduling from $\alpha_{min}$ to $\alpha_{max}$.
- **Linear** : Apply Linear scheduling from $\alpha_{min}$ to $\alpha_{max}$.
- **Step** : At each $t_{iter}$ iterations, multiply $\alpha$ by 2 until $\alpha = \alpha_{max}$.

Note that the standard cosine scheduling technique (Loshchilov & Hutter, 2017) typically works by decreasing the target parameters. In this case, we multiplied the scheduling term by $(-1)$.

**Generation Results** We tested our UOTM-SD model on CIFAR-10 ($32 \times 32$) and CelebA-HQ ($256 \times 256$) datasets. For quantitative evaluation, we adopted FID (Heusel et al., 2017) score. Tab 3 shows that our UOTM-SD improves UOTM (Choi et al., 2023a) across all three scheduling schemes. Our UOTM-SD achieves a FID of 2.51 under the best setting of linear scheduling with $(\alpha_{min}, \alpha_{max}) = (1/5, 5)$ and $\tau = 1 \times 10^{-3}$, surpassing all other OT-based methods. (See Appendix D.5 for the qualitative comparison of generated samples.) We tested UOTM-SD with linear scheduling, which performed best on CIFAR-10, on CelebA-HQ. Our UOTM-SD outperformed the previous best-performing OT-based model (UOTM) (Tab 4). We added a more extensive comparison with

other generative models in Appendix D.2. (Due to page constraints, we included the ablation study regarding schedule intensity, i.e., $(\alpha_{min}, \alpha_{max})$, and the schedule itself, i.e., $\alpha_{min} = \alpha_{max}$, in Appendix D.4.)

$\tau$ **Robustness** We assessed the robustness of our model regarding the intensity parameter $\tau$ of the cost function $c(x, y)$. Specifically, we tested whether our UOTM-SD resolves the $\tau$-sensitivity of UOTM, observed in Fig 3. Fig 5 displays FID scores of UOTM-SD, UOTM, and OTM for various values of $\tau$. Note that we employed harsh conditions for $\tau$-robustness, where $\tau_{max}/\tau_{min} = 25$. We adopted $\alpha_{min} = 1/5$ and $\alpha_{max} = 5$ for each UOTM-SD. All three versions of UOTM-SD outperform UOTM and OTM under the same $\tau$. While UOTM shows large variation of FID scores depending on $\tau$, ranging from $2.71$ to $218.02$, UOTM-SD provides much more stable results. (See Appendix D.2 for table results.)

## 5 CONCLUSION

In this paper, we integrated and analyzed various OT-based GANs. Our analysis unveiled that establishing $g_1$ and $g_2$ as lower-bounded, non-decreasing, and strictly convex functions significantly enhances training stability. Moreover, the cost function $c$ contributes to alleviating mode collapse and mixture problems. Nevertheless, UOTM, which leverages these two factors, exhibits $\tau$-sensitivity. In this regard, we suggested a novel approach that addresses this $\tau$-sensitivity of UOTM while achieving improved best-case results. However, there are some limitations to our work. Firstly, we fixed $g_3 = \text{Id}$ during our analysis. Also, our convergence theorem for $\alpha$-scaled UOT guarantees the convergence of the transport plan, but not the transport map. Exploring these issues would be promising future research.

## ACKNOWLEDGEMENTS

This work was supported by KIAS Individual Grant [AP087501] via the Center for AI and Natural Sciences at Korea Institute for Advanced Study, the NRF grant[2021R1A2C3010887], and MSIT/ IITP[NO.2021-0-01343, Artificial Intelligence Graduate School Program(SNU)].

## REPRODUCIBILITY

To ensure the reproducibility of this work, we submitted the source code in the supplementary materials. The implementation details of all experiments are clarified in Appendix B. Moreover, the assumptions and complete proofs for Theorem 3.1 and 4.1 are included in Appendix A.

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

# A PROOFS

**Notations and Assumptions**   Let $\mathcal{X}$ and $\mathcal{Y}$ be compact complete metric spaces which are convex subsets of $\mathbb{R}^d$, and $\mu$, $\nu$ be positive Radon measures of the mass 1. For a measurable map $T : \mathcal{X} \to \mathcal{Y}$, $T_{\#}\mu$ denotes the associated pushforward distribution of $\mu$. $c(x, y)$ refers to the transport cost function defined on $\mathcal{X} \times \mathcal{Y}$. We assume $\mathcal{X}, \mathcal{Y} \subset \mathbb{R}^d$ and the quadratic cost $c(x, y) = \tau\|x - y\|_2^2$, where $\tau$ is a given positive constant. Let $\Psi_1$ and $\Psi_2$ be an entropy function, i.e. $\Psi_i : \mathbb{R} \to [0, \infty]$ is a convex, lower-semi continuous, non-negative function such that $\Psi_i(1) = 0$, and $\Psi_i(x) = \infty$ for $x < 0$. Let $g_1 := \Psi^*$ and $g_2 := \Psi^*$ be a convex, differentiable, non-decreasing function defined on $\mathbb{R}$. We assume that $g_1(0) = g_2(0) = 0$ and $g_1'(0) = g_2'(0) = 1$.

**Csiszàr Divergence**   Let $\Psi$ be an entropy function. The Csiszàr divergence induced by $\Psi$ (or $\Psi$-divergence) between $\mu$ and $\nu$ is defined as follows:

$$D_{\Psi}(\mu|\nu) = \int_{\mathcal{Y}} \Psi\left(\frac{d\mu}{d\nu}\right) d\nu + \Psi'(\infty)\mu^{\perp}(\nu), \tag{14}$$

where $\mu = \frac{d\mu}{d\nu}\nu + \mu^{\perp}(\nu)$ is a Radon-Nikodym decomposition of $\mu$ with respect to $\nu$.

**Theorem A.1.** *Let $g_1$ and $g_2$ be real-valued functions that are non-decreasing, bounded below, differentiable, and strictly convex. Assuming the regularity assumptions in Appendix A, there exists a unique Lipschitz continuous optimal potential $v^{\star}$ for Eq. 7. Moreover, for the maximization objective $-\mathcal{L}_v$ of Eq. 7,*

$$\Gamma := \{v \in \mathcal{C}(\mathcal{Y}) : \mathcal{L}_v \leq 0, v^{cc} = v\}, \tag{15}$$

*is equi-bounded and equi-Lipschitz.*

*Proof.* Let

$$\mathcal{L}(v) = \int_{\mathcal{X}} g_1\left(-v^c(x)\right) d\mu(x) + \int_{\mathcal{Y}} g_2(-v(y)) d\nu(y). \tag{16}$$

Since $\mathcal{L}(0) = 0$, the infimum of $\mathcal{L}(v)$ is non-positive. Thus, $\Gamma$ is nonempty. We would like to prove that the set $\Gamma$ is equi-bounded and equi-Lipschitz, i.e., there exists a constant $L > 0$ such that for every $z \in \Gamma$, $v^c|_{\text{supp}(\mu)}$ and $v|_{\text{supp}(\nu)}$ are $L$-Lipschitz. Let $A$ be the lower bound of functions $g_1$ and $g_2$, i.e. $A \leq g_1(x)$ and $A \leq g_2(y)$ for $x \in \mathcal{X}$ and $y \in \mathcal{Y}$, respectively. Furthermore, since $\mathcal{X}$ and $\mathcal{Y}$ are compact, there exists $M > 0$ such that $c(x, y) \leq M$ for all $(x, y) \in \mathcal{X} \times \mathcal{Y}$. Then, since $g_1(x) \geq \text{Id}(x)$,

$$0 \geq \mathcal{L}(v) \geq \int_{\mathcal{X}} -\inf_{y \in \mathcal{Y}}(c(x, y) - v(y)) d\mu(x) + \int_{\mathcal{Y}} g_2(-v(y)) d\nu(y), \tag{17}$$

$$\geq -M + \underbrace{\sup_{y \in \mathcal{Y}}(v(y))}_{=:\tilde{v}} + \int_{\mathcal{Y}} g_2(-v(y)) d\nu(y) \geq -M + \tilde{v} + A, \tag{18}$$

which indicates that $v(y) \leq M - A$ for all $y \in \mathcal{Y}$. Note that $M$ nor $A$ is dependent on the choice of $v$. Thus, $v \in \Gamma$ is equi-bounded above. Moreover, by using similar logic with respect to $v^c$, we can easily prove that $v$ is also equibounded below. Consequently, by symmetricity, $v$ and $v^c$ are equi-bounded.

We now prove that there exists a uniform constant $L$ such that for every $z \in \Gamma$, $v$ is Lipschitz continuous with constant $L$. Since $v$ is bounded and $v^{cc} = v$, there exists a point $x(y)$ such that

$$v(y) = c(x(y), y) - v^c(x(y)), \tag{19}$$

and for every $\tilde{y} \in \mathcal{Y}$,

$$v(\tilde{y}) \leq c(x(y), \tilde{y}) - v^c(x(y)). \tag{20}$$

Subtracting the two previous inequalities gives,

$$v(\tilde{y}) - v(y) \leq c(x(y), \tilde{y}) - c(x(y), y). \tag{21}$$

Since $c$ is Lipschitz continuous on the compact domain $\mathcal{X} \times \mathcal{Y}$, there exists a Lipschitz constant $L$ that satisfies $|c(x(y), \tilde{y}) - c(x(y), y)| \leq L\|\tilde{y} - y\|_2$. Thus,

$$|v(\tilde{y}) - v(y)| \leq L\|\tilde{y} - y\|_2. \tag{22}$$

To sum up, $\Gamma$ is nonempty, equibounded, and equi-Lipschitz. Moreover, $\mathcal{L}(v) \geq 2A$, thus $\mathcal{L}(\Gamma)$ is lower-bounded.

Now, we would like to prove the compactness of $\Gamma$. Take any sequence $\{v_n\}_{n \in \mathbb{N}} \subset \Gamma$. Then, since $\Gamma$ is nonempty, equibounded, and equi-Lipschitz, we can obtain a uniformly convergent subsequence $\{v_{n_k}\}_{k \in \mathbb{N}} \to v$ by Arzelà-Ascoli theorem. Because $v_n(y) - \tau \|y\|_2^2$ is concave for each $v_n \in \Gamma$ from $v^{cc} = v$ (Santambrogio, 2015), $v$ is also continuous and $v(y) - \tau \|y\|_2^2$ is concave. Thus, $v$ is c-concave, i.e. $v^{cc} = v$. Now, to prove $v \in \Gamma$, we only need to prove $\mathcal{L}_v \leq 0$. Since $\{v_{n_k}\}_{k \in \mathbb{N}} \to v$ uniformly, it is easy to show that $\{v_{n_k}^c\}_{k \in \mathbb{N}} \to v^c$ uniformly. Moreover, note that $\{v_{n_k}^c\}_{k \in \mathbb{N}}$ is equibounded. By applying the dominated convergence theorem (DCT), we can easily prove that $\mathcal{L}_v \leq 0$. Thus, for any sequence of $\Gamma$, there exists a subsequence that converges to point of $\Gamma$ (Bolzano-Weierstrass property), which implies that $\Gamma$ is compact. Finally, since $\mathcal{L}(\Gamma)$ is lower-bounded, there exists a minimizer $v^\star \in \Gamma$, i.e. $\mathcal{L}(v^\star) \leq \mathcal{L}(v)$ for all $v \in \Gamma$ by compactness of $\Gamma$.

Now, we prove the uniqueness of the minimizer. Let $K > 0$ be a real value that $|v| \leq K$ and $|v^c| \leq K$ for every $v \in \Gamma$. There exists such $K$ by the equiboundedness. Now, let $\mathcal{C}_K$ denote the collection of continuous functions which are bounded by $K$. Since $g_1$ and $g_2$ are strictly convex on $[-K, K]$, the following dual minimization problem becomes strictly convex:

$$\inf_{(u,v) \in \mathcal{C}_K(\mathcal{X}) \times \mathcal{C}_K(\mathcal{Y})} \int_{\mathcal{X}} g_1\left(-u(x)\right) \, \mathrm{d}\mu(x) + \int_{\mathcal{Y}} g_2(-v(y)) \, \mathrm{d}\nu(y). \tag{23}$$

Thus, there exists at most one solution. Because there exists a solution $(v^{\star c}, v^\star)$, it is the unique solution. $\qquad\square$

**Theorem A.2.** *Assume the entropy functions $\Psi_1, \Psi_2$ are strictly convex and finite on $(0, \infty)$. Then, the optimal transport plan $\pi^{\alpha,\star}$ of the $\alpha$-scaled UOT problem $C_{ub}^\alpha(\mu, \nu)$ (Eq. 6) weakly converges to the optimal transport plan $\pi^\star$ of the OT problem $C(\mu, \nu)$ (Eq. 1) as $\alpha$ goes to infinity.*

*Proof.* Note that the $\alpha$-scaled UOT problem $C_{ub}^\alpha(\mu, \nu)$ is equivalent to setting the cost intensity $\tau \to \frac{\tau}{\alpha}$ within the cost function $c(x,y) = \tau \|x - y\|_2^2$ of the standard UOT problem $C_{ub}(\mu, \nu)$:

$$\pi^{\alpha,\star} = \arg\inf_{\pi^\alpha \in \mathcal{M}_+(\mathcal{X} \times \mathcal{Y})} \left[ \int_{\mathcal{X} \times \mathcal{Y}} \tau \|x - y\|_2^2 \, \mathrm{d}\pi^\alpha(x,y) + \alpha D_{\Psi_1}(\pi_0^\alpha | \mu) + \alpha D_{\Psi_2}(\pi_1^\alpha | \nu) \right],$$
$$\tag{24}$$

$$= \arg\inf_{\pi \in \mathcal{M}_+(\mathcal{X} \times \mathcal{Y})} \left[ \int_{\mathcal{X} \times \mathcal{Y}} \frac{\tau}{\alpha} \|x - y\|_2^2 \, \mathrm{d}\pi(x,y) + D_{\Psi_1}(\pi_0 | \mu) + D_{\Psi_2}(\pi_1 | \nu) \right]. \tag{25}$$

Choi et al. (2023a) proved that, in the standard UOT problem, the marginal discrepancies for the optimal $\pi^\star$ are linearly proportional to the cost intensity. This relationship can be interpreted as follows for the above $\pi^{\alpha,\star}$:

$$D_{\Psi_1}(\pi_0^{\alpha,\star} | \mu) + D_{\Psi_2}(\pi_1^{\alpha,\star} | \nu) \leq \frac{\tau}{\alpha} \mathcal{W}_2^2(\mu, \nu). \tag{26}$$

Therefore, as $\alpha$ goes to infinity, the marginal distributions of $\pi^{\alpha,\star}$ converge in the Csiszàr divergences to the source $\mu$ and target $\nu$ distributions:

$$\lim_{\alpha \to \infty} D_{\Psi_1}(\pi_0^{\alpha,\star} | \mu) = \lim_{\alpha \to \infty} D_{\Psi_2}(\pi_1^{\alpha,\star} | \nu) = 0. \tag{27}$$

The convergence in Csiszar divergence $D_{\Psi_i}$ for a strictly convex $\Psi_i$ implies the convergence of measures in Total Variation distance (Sason & Verdú, 2016; Csiszár, 1972). Then, this convergence in Total Variation distance implies the weak convergence of measures. This can be easily shown as follows: For any continuous and bounded $f \in \mathcal{C}_b(\mathcal{X})$, we have

$$\left| \int f d\mu_n - \int f d\mu \right| = \left| \int f d(\mu_n - \mu) \right| = \|f\|_\infty \left| \int (f/\|f\|) \mathrm{d}(\mu_n - \mu) \right|, \tag{28}$$

$$\leq \|f\|_\infty \|\mu_n - \mu\|_{TV}. \tag{29}$$

Therefore, $\pi_0^{\alpha,\star}$ and $\pi_1^{\alpha,\star}$ weakly converges to $\mu$ and $\nu$, respectively. Choi et al. (2023a) showed that the optimal $\pi^{\alpha,\star}$ of Eq. 25 becomes the optimal transport plan for the OT problem $C(\pi_0^{\alpha,\star}, \pi_1^{\alpha,\star})$ for

the same cost function $c(x, y) = \tau \|x - y\|_2^2$. (The optimal transport plan is invariant to the constant scaling of the cost function). Moreover, since $\mathcal{X}, \mathcal{Y}$ are compact, $c(x, y)$ is bound on $\mathcal{X} \times \mathcal{Y}$. Thus,

$$\liminf_{\alpha \to \infty} \int c(x, y) \, \mathrm{d}\pi^{\alpha,\star} < \infty. \tag{30}$$

Consequently, Theorem 5.20 from Villani et al. (2009) proves that $\pi^{\alpha,\star}$ weakly converges to the optimal transport plan $\pi^\star$ of the OT problem $C(\mu, \nu)$ as $\alpha$ goes to infinity.

$\square$

## B  IMPLEMENTATION DETAILS

For every implementation, the prior (source) distribution is a standard Gaussian distribution with the same dimension as the data (target) distribution.

**2D Experiments**  For $m_i = 12 \left( \cos \frac{i}{4}\pi, \sin \frac{i}{4}\pi \right)$ for $i = 0, 1, \ldots, 7$ and $\sigma = 0.4$, we set mixture of $\mathcal{N}(m_i, \sigma^2)$ a target distribution. For all synthetic experiments, we used the same generator and discriminator network architectures. The auxiliary variable $z$ has a dimension of two. For a generator, we passed $z$ through two fully connected (FC) layers with a hidden dimension of 128, resulting in 128-dimensional embedding. We also embedded data $x$ into the 128-dimensional vector by passing it through three-layered ResidualBlock (Song & Ermon, 2019). Then, we summed up the two vectors and fed them to the final output module. The output module consisted of two FC layers. For the discriminator, we used three layers of ResidualBlock and two FC layers (for the output module). The hidden dimension is 128. Note that the SiLU activation function is used. We used a batch size of 128, and a learning rate of $2 \times 10^{-4}$ and $10^{-4}$ for the generator and discriminator, respectively. We trained for 30K iterations. For OTM and UOTM, we chose the best results between settings of $\tau = 0.01, 0.05$. OTM has shown the best performance with $\tau = 0.05$ and UOTM has shown the best performance with $\tau = 0.01$. For WGANs and OTM, since they do not converge without any regularization, we set the regularization parameter $\lambda = 5$. We used a gradient clip of $0.1$ for WGAN.

**CIFAR-10**  For the DCGAN model, we employed the architecture of Balaji et al. (2020), which uses convolutional layers with residual connection. Note that this is the same model architecture as in Rout et al. (2022); Choi et al. (2023a). We set a batch size of 128, 50K iterations, a learning rate of $2 \times 10^{-4}$ and $10^{-4}$ for the generator and discriminator, respectively. In the DCGAN backbone, we adopt a simple practical scheme suggested in OTM Rout et al. (2022) for accommodating a smaller dimension for the input latent space $X$. This practical scheme involves introducing a deterministic bicubic upsampling $Q$ from $\mathcal{X}$ to $\mathcal{Y}$. Then, we consider the OT map between $Q_\# \mu$ and $\nu$. In practice, we sample $x$ in Algorithm 1 from a 192-dimensional standard Gaussian distribution. Then, $x$ is directly used as an input for the DCGAN generator $T_\theta$. The random variable $z$ is not employed in the DCGAN implementation. Meanwhile, $Q(x)$ is obtained by reshaping $x$ into a $3 \times 8 \times 8$ dimensional tensor, and then bicubically upsampling it to match the shape of the image. The generator loss is defined as $c(Q(x), T_\theta(x)) - v_\phi(T_\theta(x))$.

For the NCSN++ model, we followed the implementation of Choi et al. (2023b) unless otherwise stated. Specifically, we set $\mathcal{X} = \mathcal{Y}$ and use $c(x, y) = \tau \|x - y\|^2$ without introducing upsampling $Q$. Here, the auxiliary variable $z$ is employed. We sample $z$ from a 256-dimensional Gaussian distribution and put it as an additional stochastic input to the generator. The input prior sample $x$ is fed into the NCSN++ network like UNet input. The auxiliary $z$ passes through embedding layers and is incorporated into the intermediate feature maps of the NCSN++ through an attention module. We trained for 200K for OTM and 120K for other models because OTM converges slower than other models. Moreover, we used $R_1$ regularization of $\lambda = 0.2$ for all methods and architectures. WGANs are known to show better performance with the optimizers without momentum term, thus, we use Adam optimizer with $\beta_1 = 0$, for WGANs. Furthermore, since OTM has a similar algorithm to WGAN, we also use Adam optimizer with $\beta_1 = 0$. Lastly, following Choi et al. (2023a), we use Adam optimizer with $\beta_1 = 0.5$ for UOTM. Note that for all experiments, we use $\beta_2 = 0.9$ for the optimizer. We used a gradient clip of $0.1$ for WGAN. Furthermore, the implementation of UOTM-SD follows the UOTM hyperparameter unless otherwise stated. We trained UOTM-SD for 200K iterations. For UOTM-SD (Cosine) and (Linear), we initiated the scheduling strategy from the

start and finished the scheduling at 150K iterations. For UOTM-SD (Step), we halved $\tilde{\alpha}$ for every 30K iterations until it reaches $\alpha_{max}^{-1}$.

**Evaluation Metric**    For the evaluation of image datasets, we used 50,000 generated samples to measure FID (Karras et al., 2018) scores. For every model, we evaluate the FID score for every 10K iterations and report the best score among them.

## C    PROBLEMS OF GAN-BASED GENERATIVE MODELS

**Unstable training**    Training adversarial networks involves finding a Nash equilibrium (Osborne & Rubinstein, 1994) in a two-player non-cooperative game, where each player aims to minimize their own objective function. However, discovering a Nash equilibrium is an exceedingly challenging task (Salimans et al., 2016; Mescheder et al., 2018). The prevailing approach for adversarial training is to adopt alternating gradient descent updates for the generator and discriminator. Unfortunately, the gradient descent algorithm often struggles to converge for many GANs (Salimans et al., 2016; Mescheder et al., 2018). Notably, Mescheder et al. (2018) showed that neither WGANs nor WGANs with Gradient Penalty (WGAN-GP) offer stable convergence.

**Mode collapse/mixture**    Another primary challenge in adversarial training is mode collapse and mixture phenomena. Mode collapse means that a generative model fails to encompass all modes of the data distribution. Conversely, mode mixture represents that a generative model fails to separate two modes of data distribution while attempting to cover all modes. This results in the generation of spurious or ambiguous samples. Many state-of-the-art GANs enforce regularization on the spectral norm (Miyato et al., 2018) to mitigate training instability. Odena et al. (2018) showed that enforcing the magnitude of the spectral norm of the networks reduces instability in training. However, recent works (Nagarajan & Kolter, 2017; Khayatkhoei et al., 2018; An et al., 2020a; Salmona et al., 2022) have revealed that such Lipschitz constraints can lead the generator to concentrate solely on one of the modes or lead to mode mixtures in the generated samples.

## D    ADDITIONAL RESULTS

### D.1    ADDITIONAL QUALITATIVE RESULTS ON TOY DATASETS

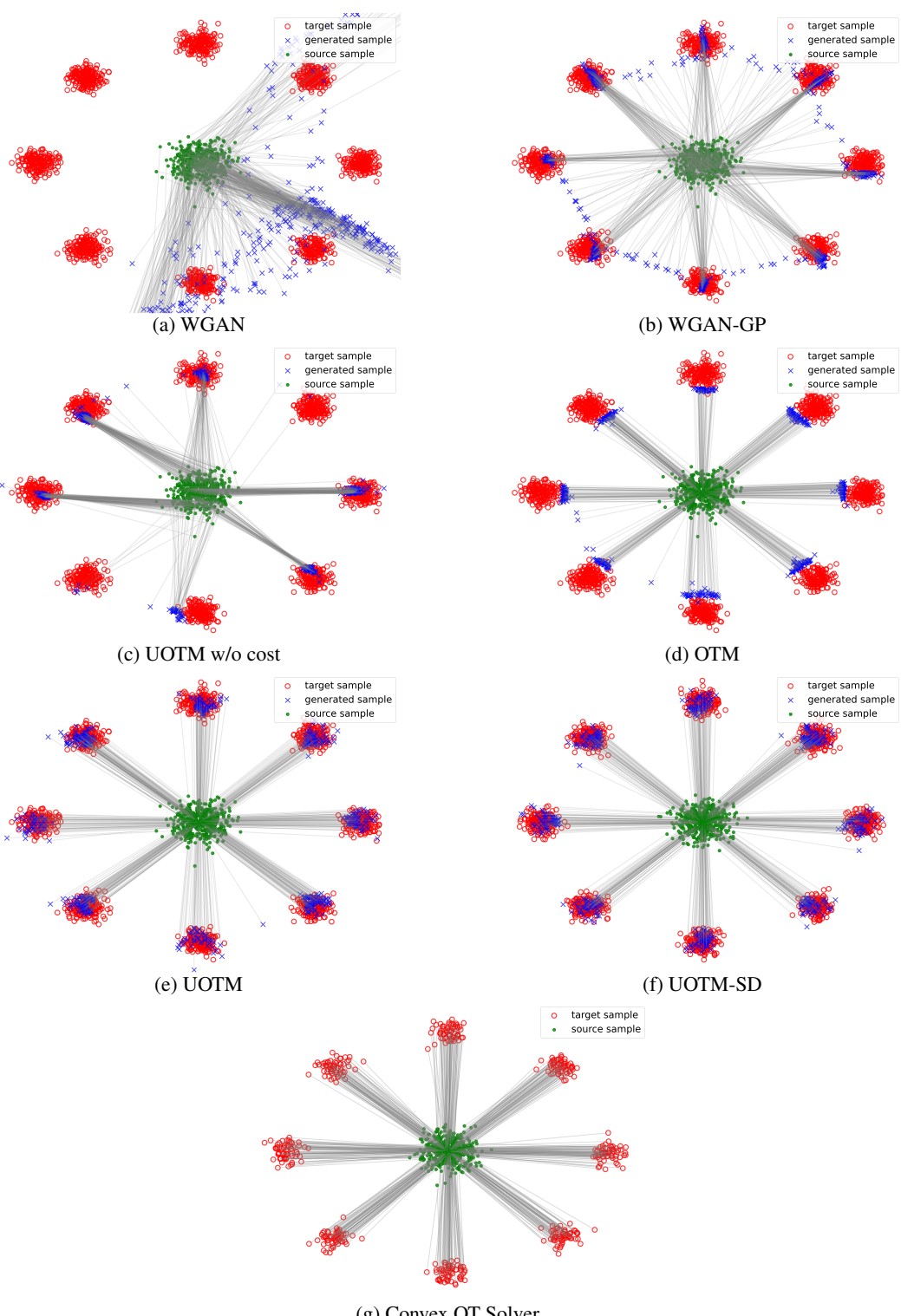

Figure 7: **Visualization of Generator** $T_\theta$. The gray lines illustrate the generated pairs, i.e., the connecting lines between $x$ (green) and $T_\theta(x)$ (blue). The red dots represent the training data samples.

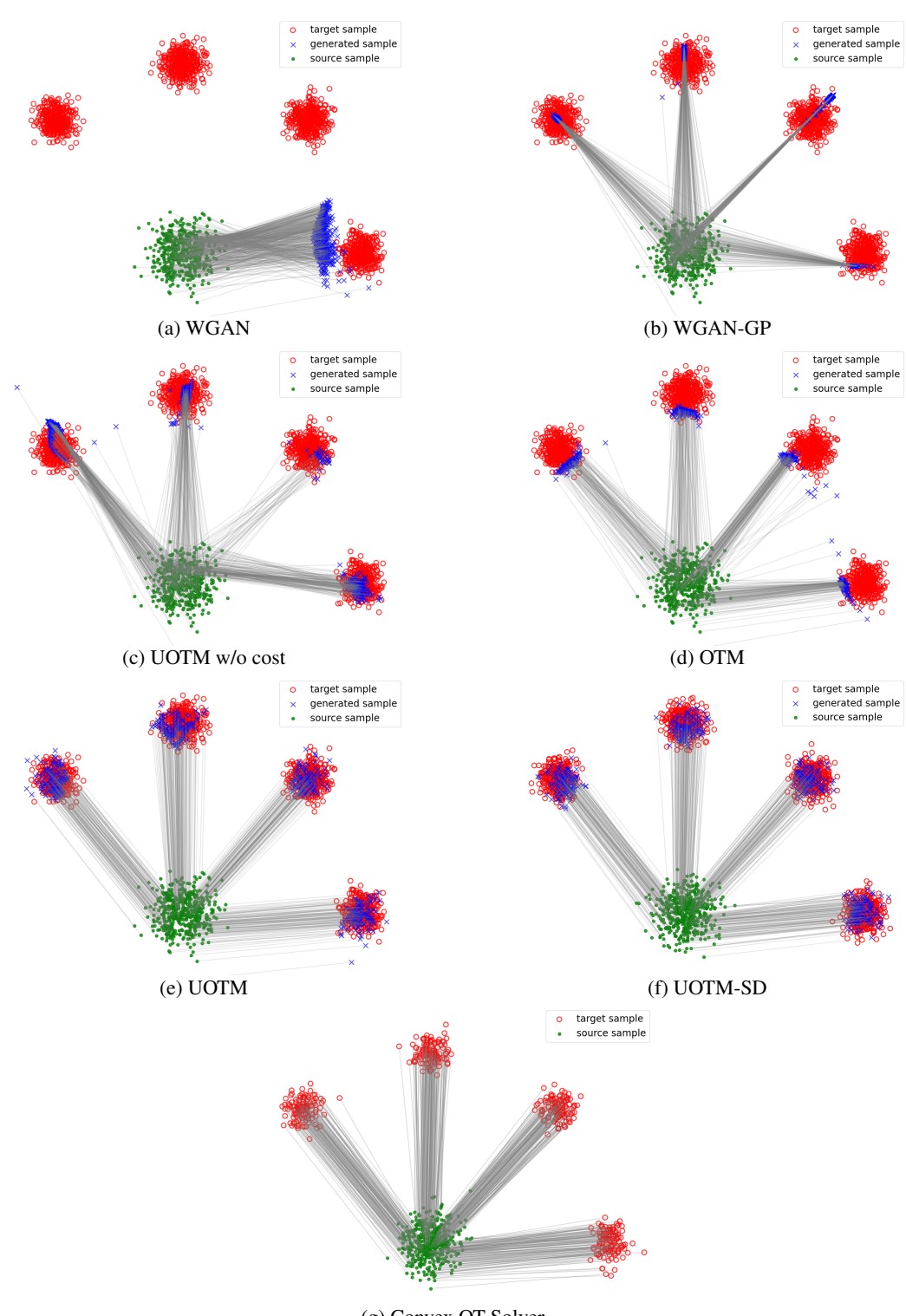

Figure 8: **Visualization of Generator** $T_\theta$**.** The gray lines illustrate the generated pairs, i.e., the connecting lines between $x$ (green) and $T_\theta(x)$ (blue). The red dots represent the training data samples.

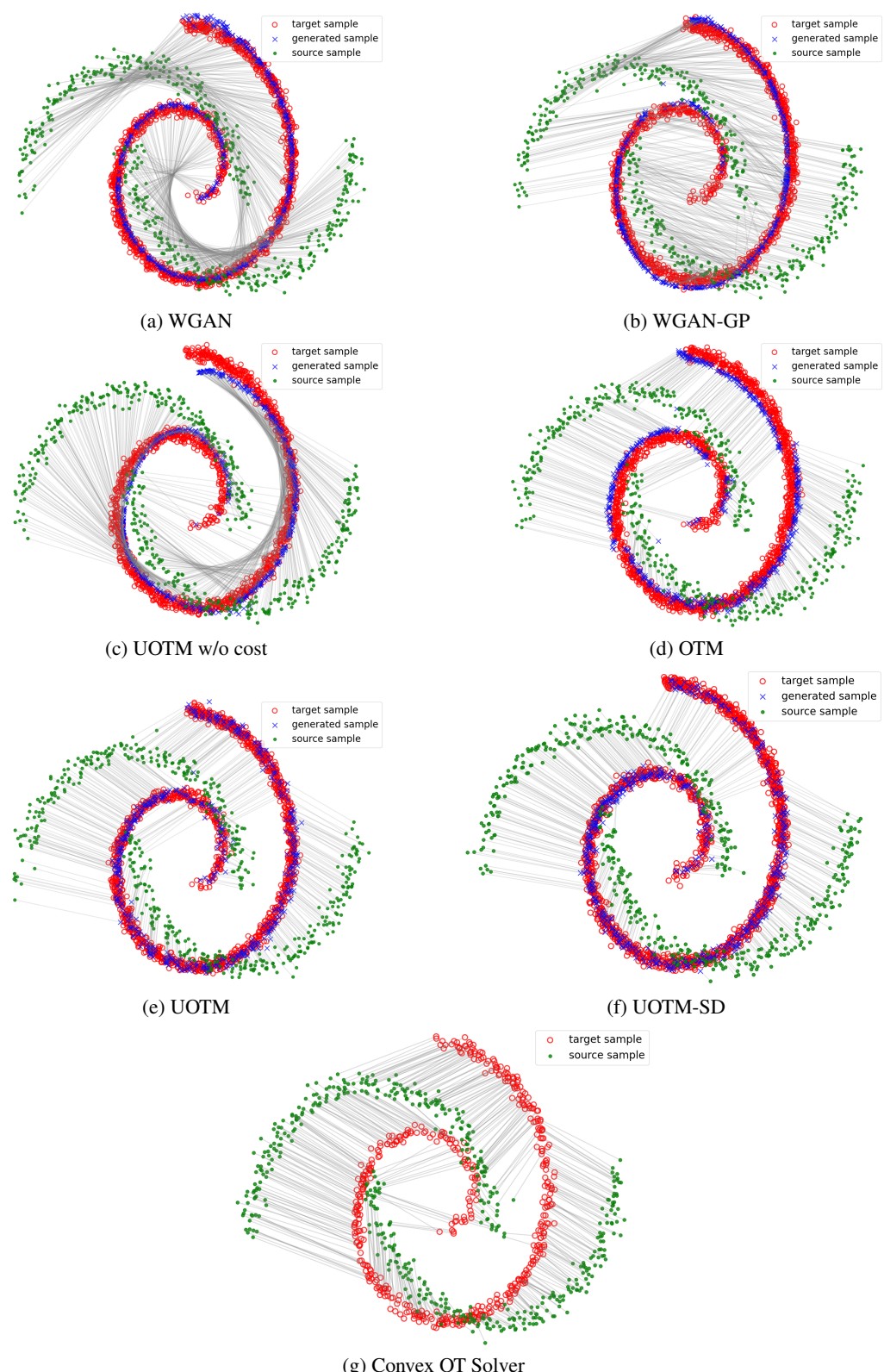

Figure 9: **Visualization of Generator** $T_\theta$. The gray lines illustrate the generated pairs, i.e., the connecting lines between $x$ (green) and $T_\theta(x)$ (blue). The red dots represent the training data samples.

### D.2 FULL TABLE RESULT FOR CIFAR-10 GENERATION

Table 6: **Quantitative Evaluation of OT-based GANs** on CIFAR-10.

| Model | Backbone Architecture | | | |
| | NCSN++ | | | DCGAN |
| | FID ($\downarrow$) | Precision ($\uparrow$) | Recall ($\uparrow$) | FID ($\downarrow$) |
|---|---|---|---|---|
| WGAN | 48.8 | 0.45 | 0.02 | 52.3 |
| WGAN-GP | 4.5 | 0.71 | 0.55 | 50.8 |
| OTM | 4.3 | 0.71 | 0.49 | 19.8 |
| UOTM w/o cost | 19.7 | **0.80** | 0.13 | 15.4 |
| UOTM (SP) | **2.7** | 0.78 | **0.62** | 15.8 |
| UOTM (KL) | 2.9 | - | - | **12.2** |

Table 7: **Comparison of $\tau$-robustness.** We use $\alpha_{min} = 1/5$ and $\alpha_{max} = 5$ for each UOTM-SD.

| $\tau$ | 2e-4 | 5e-4 | 1e-3 | 2e-3 | 5e-3 |
|---|---|---|---|---|---|
| UOTM-SD (Cosine) | **3.60** | 2.99 | 2.57 | 2.95 | 5.42 |
| UOTM-SD (Linear) | 4.18 | 3.01 | **2.51** | 3.39 | **4.62** |
| UOTM-SD (Step) | 3.92 | **2.81** | 2.78 | **2.89** | 5.34 |
| UOTM | 15.19 | 22.02 | 2.71 | 6.30 | 218.02 |
| OTM | 4.34 | 4.15 | 4.38 | 5.13 | 7.43 |

Table 8: **Image Generation on CIFAR-10.** † indicates the results conducted by ourselves.

| Class | Model | FID (↓) |
|---|---|---|
| **GAN** | SNGAN+DGflow (Ansari et al., 2020) | 9.62 |
| | AutoGAN (Gong et al., 2019) | 12.4 |
| | TransGAN (Jiang et al., 2021) | 9.26 |
| | StyleGAN2 w/o ADA (Karras et al., 2020) | 8.32 |
| | StyleGAN2 w/ ADA (Karras et al., 2020) | 2.92 |
| | DDGAN (T=1)(Xiao et al., 2021) | 16.68 |
| | DDGAN (Xiao et al., 2021) | 3.75 |
| | RGM (Choi et al., 2023b) | **2.47** |
| **Diffusion** | NCSN (Song & Ermon, 2019) | 25.3 |
| | DDPM (Ho et al., 2020) | 3.21 |
| | Score SDE (VE) (Song et al., 2021b) | 2.20 |
| | Score SDE (VP) (Song et al., 2021b) | 2.41 |
| | DDIM (50 steps) (Song et al., 2021a) | 4.67 |
| | CLD (Dockhorn et al., 2022) | 2.25 |
| | Subspace Diffusion (Jing et al., 2022) | 2.17 |
| | LSGM (Vahdat et al., 2021) | **2.10** |
| **VAE&EBM** | NVAE (Vahdat & Kautz, 2020) | 23.5 |
| | Glow (Kingma & Dhariwal, 2018) | 48.9 |
| | PixelCNN (Van Oord et al., 2016) | 65.9 |
| | VAEBM (Xiao et al., 2020) | 12.2 |
| | Recovery EBM (Gao et al., 2021) | **9.58** |
| **OT-based** | WGAN (Arjovsky et al., 2017) | 55.20 |
| | WGAN-GP(Gulrajani et al., 2017) | 39.40 |
| | Robust-OT (Balaji et al., 2020) | 21.57 |
| | AE-OT-GAN (An et al., 2020b) | 17.10 |
| | OTM† (Rout et al., 2022) | 4.15 |
| | UOTM (Choi et al., 2023a) | 2.97 |
| | UOTM-SD (Cosine)† | 2.57 |
| | UOTM-SD (Linear)† | **2.51** |
| | UOTM-SD (Step)† | 2.78 |

## D.3 ADDITIONAL QUANTITATIVE RESULTS FOR LIPSCHITZNESS OF POTENTIAL

Table 9: **Image Generation on CelebA-HQ.** † indicates the results conducted by ourselves.

| Class | Model | FID ($\downarrow$) |
|---|---|---|
| **Diffusion** | Score SDE (VP) Song et al. (2021b) | 7.23 |
| | Probability Flow Song et al. (2021b) | 128.13 |
| | LSGM Vahdat et al. (2021) | 7.22 |
| | UDM Kim et al. (2021) | 7.16 |
| | DDGAN Xiao et al. (2021) | 7.64 |
| | RGM Choi et al. (2023b) | **7.15** |
| **GAN** | PGGAN Karras et al. (2017) | 8.03 |
| | Adv. LAE Pidhorskyi et al. (2020) | 19.2 |
| | VQ-GAN Esser et al. (2021) | 10.2 |
| | DC-AE Parmar et al. (2021) | 15.8 |
| | StyleSwin (Zhang et al., 2022) | **3.25** |
| **VAE** | NVAE Vahdat & Kautz (2020) | 29.7 |
| | NCP-VAE Aneja et al. (2021) | 24.8 |
| | VAEBM Xiao et al. (2020) | **20.4** |
| **OT-based** | OTM† (Rout et al., 2022) | 13.56 |
| | UOTM (KL) (Choi et al., 2023a) | 6.36 |
| | UOTM (SP)† | 6.31 |
| | **UOTM-SD†** | **5.99** |

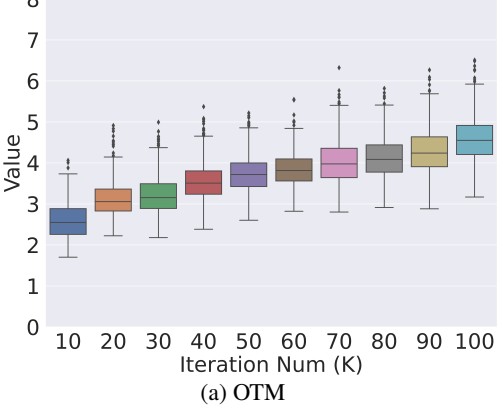
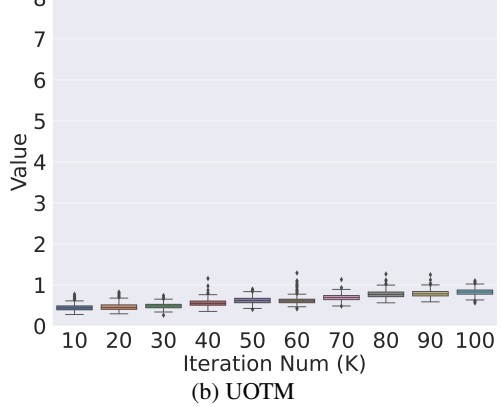

Figure 10: **Distribution of the norm of the potential gradient** $\|\nabla_y v_\phi(y)\|, \|\nabla_{\hat{y}} v_\phi(\hat{y})\|$ at a random real data $y$ and a randomly generated data $\hat{y}$ for every 10K iterations on CIFAR-10. Due to the equi-Lipschitz property, the gradient norm of UOTM potential is stable during training. This stability contributes to the stable training of UOTM. In the Toy dataset, we measured the Average Rate of Change (ARC) of potential $\frac{|v_\phi(y) - v_\phi(x)|}{\|y-x\|}$ between a randomly selected training data $x$ and another randomly chosen point $y$ within the data space (Fig 6). However, unlike the Toy dataset, the image dataset is extremely sparse in its ambient space (pixel space). Hence, randomly selecting point $y$ within the pixel space can yield undesirable results. Therefore, instead of measuring the Average Rate of Change (ARC) of potential, we measured the norm of the potential gradient.

### D.4 ADDITIONAL DISCUSSIONS ON SCHEDULING

**Schedule Intensity Ablation** To analyze the effect of schedule intensity further, we evaluated our UOTM-SD model for four different scheduling intensities. For simplicity, we focused on symmetric ones i.e., $\alpha_{max} = k, \alpha_{min} = 1/k$ for some $k > 1$, while fixing $\tau = 0.001$. Overall, the Linear Scheduling scheme provided the best result, achieving FID scores below 3 for all scheduling intensities. Nevertheless, the other two scheduling schemes also demonstrated robust performance.

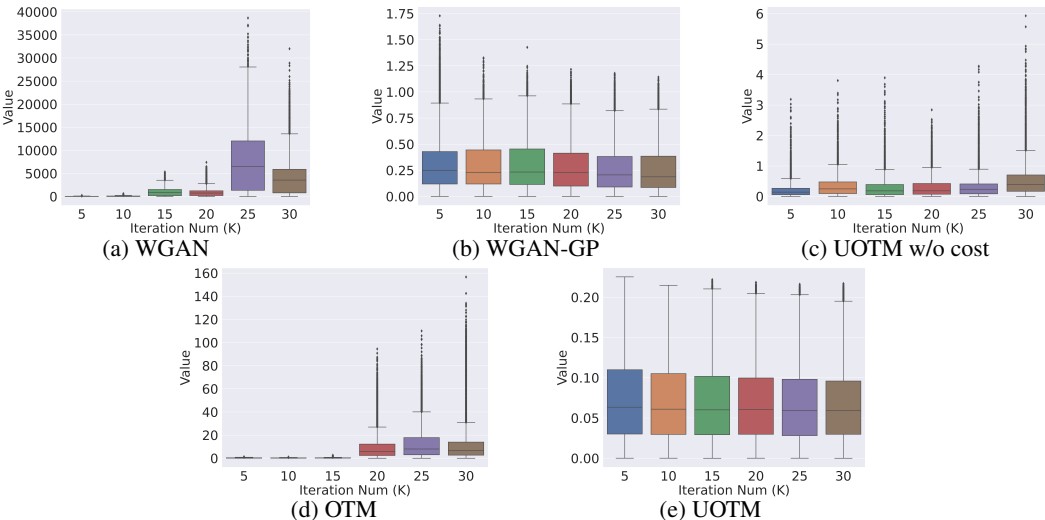

Figure 11: **Distribution of the absolute value of Average Rate of Change (ARC) of potential** $\frac{|v_\phi(y) - v_\phi(x)|}{\|y-x\|}$ for every 5K iterations. Due to the equi-Lipschitz property, |ARC| of UOTM potential is stable during training. This stability contributes to the stable training of UOTM.

Table 10: **Ablation Study on Schedule Intensity** $(\alpha_{min}, \alpha_{max})$.

| Schedule Type | (1/2, 2) | (1/3, 3) | (1/5, 5) | (1/10, 10) | (3, 3) | (5, 5) |
|---|---|---|---|---|---|---|
| Cosine | 3.20 | 2.94 | 2.57 | 2.78 | | |
| Linear | **2.70** | 2.97 | **2.51** | 2.77 | 3.73 | 3.99 |
| Step | 3.29 | **2.85** | 2.78 | **2.70** | | |

Moreover, we tested UOTM-SD without scheduling ($\alpha$-UOTM). Specifically, we tested the setting of $\alpha_{max} = \alpha_{min} = \alpha_{const} > 1$. In this case, the divergence weight $\alpha$ in Eq. is constant throughout training. When we set $\alpha_{const} = 3, 5$, $\alpha$-UOTM showed FID scores of 3.73 and 3.99. This result demonstrates that $\alpha$-scheduling provides a method for harnessing the advantages of both the large $\tau$ regime and the small $\tau$ regime. Therefore, UOTM-SD outperforms $\alpha$-UOTM.

## D.5 Additional Qualitative Results

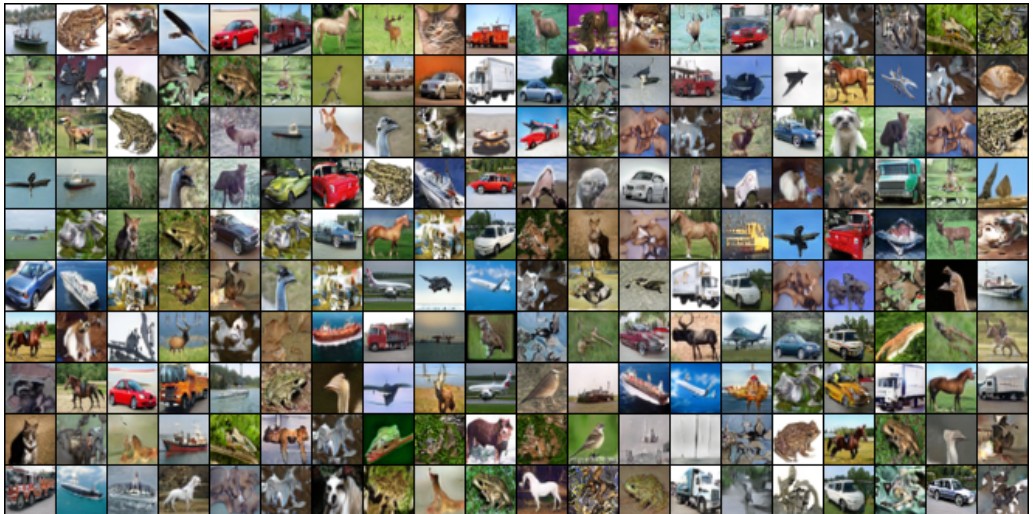

Figure 12: **Generated samples from UOTM with Small** $\tau (= 0.0002)$ on CIFAR-10 ($32 \times 32$).

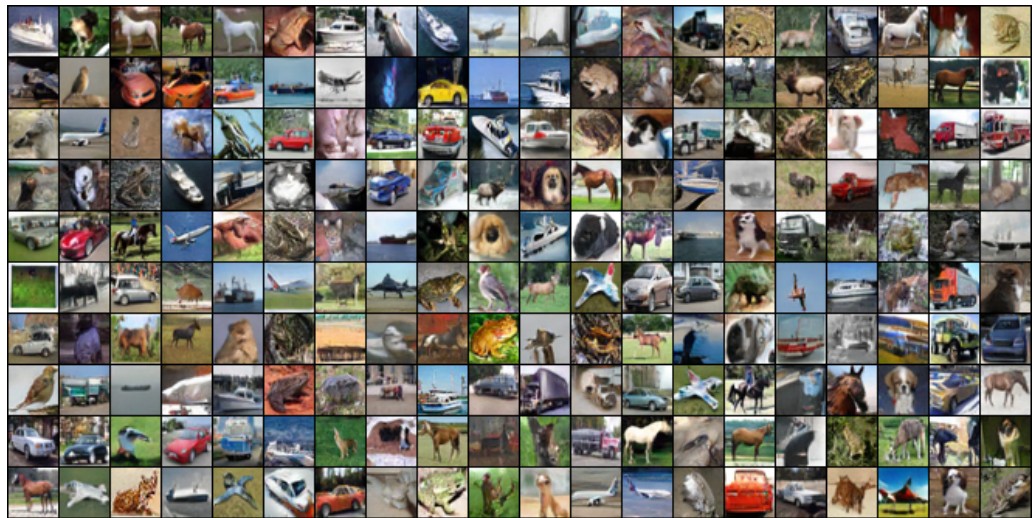

Figure 13: **Generated samples from UOTM with Optimal** $\tau(= 0.001)$ on CIFAR-10 ($32 \times 32$).

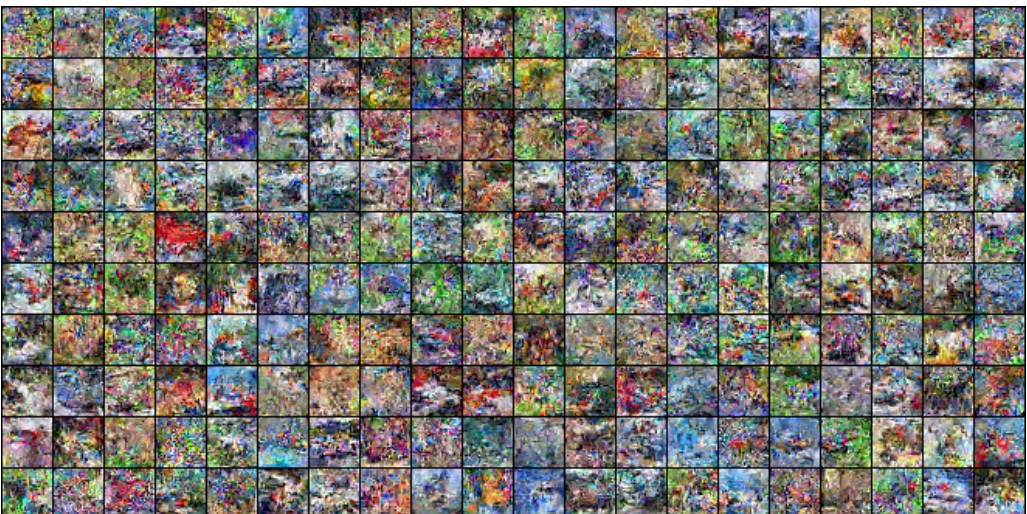

Figure 14: **Generated samples from UOTM with Large** $\tau(= 0.005)$ on CIFAR-10 ($32 \times 32$).

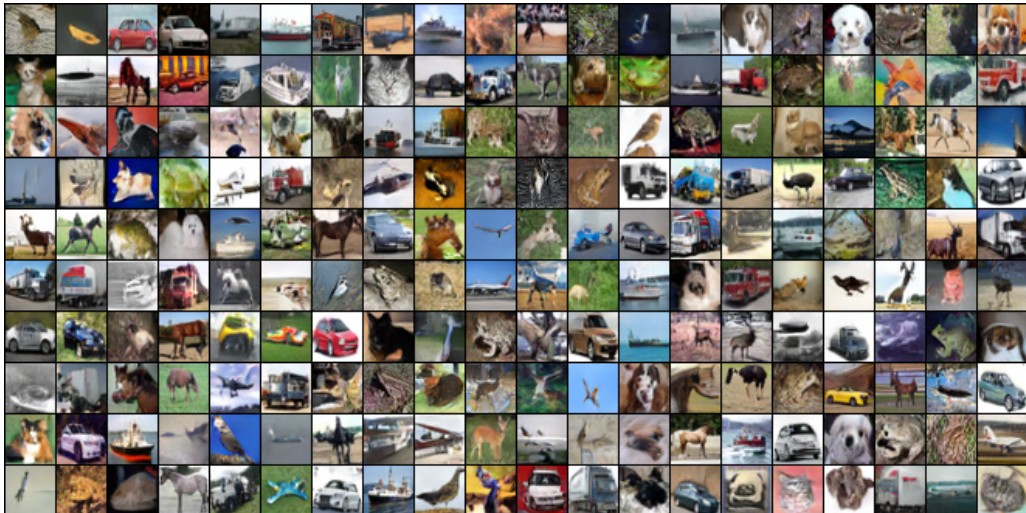

Figure 15: **Generated samples from UOTM-SD (Cosine)** on CIFAR-10 ($32 \times 32$).

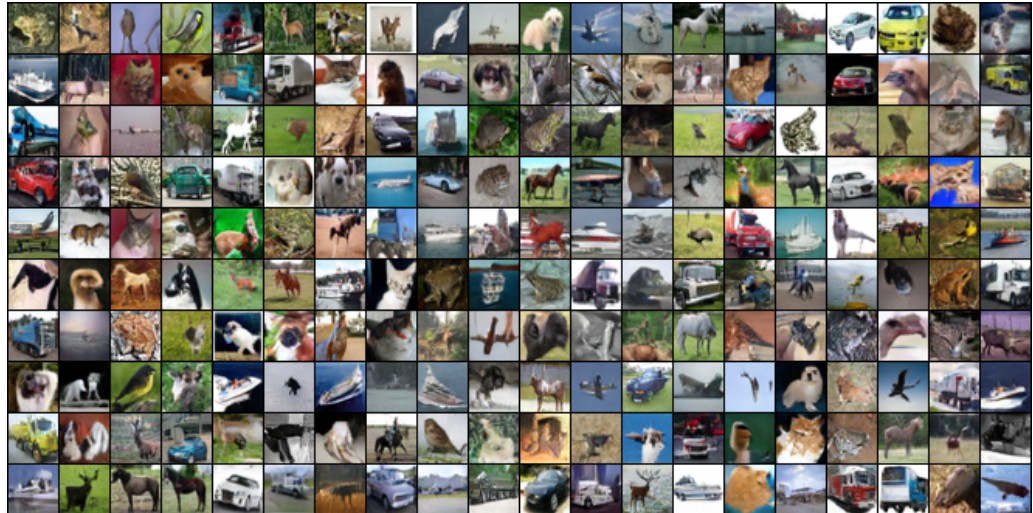

Figure 16: **Generated samples from UOTM-SD (Linear)** on CIFAR-10 ($32 \times 32$).

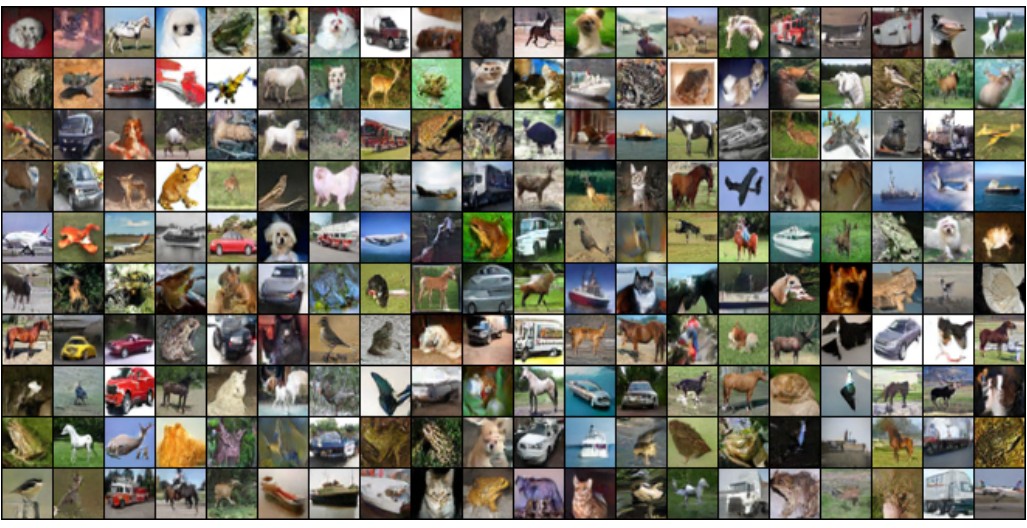

Figure 17: **Generated samples from UOTM-SD (Step)** on CIFAR-10 ($32 \times 32$).

