# OpenReview forum: "Analyzing and Improving Optimal-Transport-based Adversarial Networks"
_ICLR.cc/2024/Conference — ICLR 2024 poster_

### Official Review · Reviewer_h5iC · 2023-10-25

**Soundness:** 3 good
**Presentation:** 4 excellent
**Contribution:** 3 good
**Rating:** 6
**Confidence:** 4

**Summary:**

This paper introduced a unified framework that encompasses previous OT-based GANs, which is derived from the semi-dual form of unbalanced OT.  The authors presented a comprehensive analysis of different OT-based frameworks, as well as other well-researched generative models such as diffusion and VAEs.  In the end, the authors demonstrated a tradeoff between perception quality and mode collapsing, which was affected by the effect of the cost term in the loss function, and proposed a new training scheme by controlling this term during training.

**Strengths:**

This is an extremely well-written paper.  The presentation is easy to follow and technical details are clear and sound.

The findings in the paper are interesting.  Although the newly proposed scheduling scheme isn't the most impressive, but I trust the insight of this unified view will deepen the understanding on this topic.

The experiments are well set up to prove the hypothesis.  In general, I enjoyed reading the paper very much.

**Weaknesses:**

The main weakness of this paper is the proposed scheduling scheme is a very simple idea which proved to be useful for a basic experiment in an extremely low dimensional setting.  I'm not sure whether this method can be scaled to high dimensional setting easily.

**Questions:**

I'm surprised that WGAN didn't even work at all for the first toy problem.  How many hyperparameter settings did you try?  What would be the reason you think that it just won't work?

---

> ### Author Response · Authors · 2023-11-17
> **Response to Reviewer h5iC**
>
> We sincerely appreciate the reviewer for acknowledging that "the insight of this unified view will deepen the understanding on this topic.". Also, we are deeply grateful to the reviewer for reading our paper and offering valuable feedback.
>
> $ $
>
> ---
> **W1.**
> The main weakness of this paper is the proposed scheduling scheme is a very simple idea which proved to be useful for a basic experiment in an extremely low dimensional setting. I'm not sure whether this method can be scaled to high dimensional setting easily.
>
> **A1.**
> We evaluated our UOTM-SD model on the high-resolution dataset, **CelebA-HQ ($256 \times 256$)**. Due to limitations in time and resources, we conducted one experiment for each model with NCSN++ backbone: UOTM-SD with Linear scheduling ($g_1=g_2=SP$), UOTM ($g_1=g_2=SP$), and OTM. We selected these three models as they represent competitive options among OT-based adversarial networks on CIFAR-10 (Table 3). The FID scores below demonstrate our UOTM-SD model outperforms the other two OT map models.
>
> |Model|FID ($\downarrow$)|
> |:---|:---|
> |OTM|13.56|
> |UOTM (SP)|9.78|
> |UOTM-SD (Linear)|**8.19**|
>
> The training hyperparameters are as follows: the cost intensity $\tau=0.00001$, R1 regularization intensity $\lambda=5$, Scheduling intensity $(\alpha_{min}, \alpha_{max})=(1/5, 5)$.
>
>
> $ $
>
> ---
> **Q1.**
> I'm surprised that WGAN didn't even work at all for the first toy problem. How many hyperparameter settings did you try? What would be the reason you think that it just won't work?
>
> **A2.**
> We conducted experiments on WGAN with various hyperparameters, including learning rates (1e-4, 2e-4), weight-clipping parameters (0.01, 0.1, 1), and $R_1$ penalization term ($\lambda=0,1,5$). Unfortunately, the model did not exhibit satisfactory convergence. In response to the comment from the reviewer 12Gz, **we conducted experiments on other Toy datasets: Uneven Gaussian Mixture and Moon-to-Spiral. In these datasets, WGAN showed good performance in Fig 10.** The Toy dataset in Fig 5 requires challenging mode coverage for the generative model. We believe this is why WGAN is not successful in Fig 5.

---

> > ### Author Response · Authors · 2023-11-22
> >
> > We thank the reviewer for the efforts in reviewing our paper. We would appreciate it if the reviewer let us know whether our response was helpful in addressing the reviewer's concerns. If there are additional concerns or questions, please let us know.

---

### Official Review · Reviewer_1bYC · 2023-10-30

**Soundness:** 3 good
**Presentation:** 3 good
**Contribution:** 3 good
**Rating:** 6
**Confidence:** 3

**Summary:**

In this paper, the authors trying to unify various descriptions of OT-based adversarial networks, and  compare known frameworks under the proposed unified framework. In their unified framework, i.e. Algorithm 1, there are some degrees of freedom:
- functions $g_{1,2,3}$, that measure each term of the adversarial loss:
    - $g_1$ is for discriminator's loss from c-transform,
    - $g_2$ is for discriminator's loss from dual potential in OT,
    - $g_3$ is for generator's loss from c-transform,
- cost function $c(x,y)$ in the context of OT:
    - $\tau$: coefficient of squared transport cost $c(x,y) = \tau \|x-y\|^2_2$,
- regularization term $\mathcal{R}$.

The choice of each component corresponds a certain generative model training protocol.

They prove some theoretical guarantees on the training protocols (Thm 3.1 and Thm 4.1) and verify the assumptions for the theorems, i.e. strictly convexity and finiteness of $\Psi$s do improve the training procedure.

In addition, they clarify the limitations on their method in 5 CONCLUSION.

**Strengths:**

- The paper is well written, and there are some theoretical results (Thm 3.1 and Thm 4.1).
- They conducted various experiments and show the results on UOTM(-SD), which is the proposed unified framework, can achieve the best performance on CIFAR-10 generation except for SOTA diffusion model (Table 3). In particular, they proposed a concrete method for preventing mode collapse problem in Section 4, by considering scheduled scaling of the objectives, i.e. UOTM-SD.

**Weaknesses:**

- I think it is slightly misleading that using $\Psi$s in eq.(8) that would correspond to $g$s in Algorithm 1.
> After discussion to the authors, I concluded that this issue will appear to be less of a problem, and I raised Soundness a little.
- On the experiments for Lipschitz continuity, Fig 6, the experiments seem to be conducted only with 2d data, and I prefer counterparts of them in training for image generations also.
- UOTM-SD look working very good in CIFAR-10, but it is not evident that is also good for other domains.

**Questions:**

Question
- Do $g_{1,2}$ in Algorithm 1 correspond $\Psi_{1,2}$ in eq.(8)? If so, why the authors change its notation?
> It has been answered by authors
- The author achieved best performance of UOTM-SD on CIFAR-10, it would be good. But how about different image data? Does the scheduling strategy also improve stability using other image data?
> It has been answered by authors

---

> ### Author Response · Authors · 2023-11-17
> **Response to Reviewer 1bYC**
>
> We are deeply grateful to the reviewer for reading our paper and offering valuable feedback.
>
> $ $
>
> ---
> **W1.**
> I think it is slightly misleading that using $\Psi$s in eq.(8) that would correspond to $g$ in Algorithm 1.
>
> **Q1.**
> Do $g_{1,2}$ in Algorithm 1 correspond $\Psi_{1,2}$ in eq.(8)? If so, why the authors change its notation?
>
> **A1.**
> $g_{1,2}$ in Algorithm 1 correspond to the $\Psi_{1,2}^{\*}$, which represents the **convex conjugate** of $\Psi\_{1,2}$, i.e., $\Psi\_{i}^{\*}(y) = \sup\_{x \in \mathbb{R}}\{\langle x, y \rangle - f(x)\}$ for $\Psi\_{i}:\mathbb{R}\rightarrow [-\infty, \infty]$. Hence, for brevity, we changed the notation from $\Psi_{1,2}^{*}$ to $g_{1,2}$.
>
>
> $ $
>
> ---
> **W2.**
> On the experiments for Lipschitz continuity, Fig 6, the experiments seem to be conducted only with 2d data, and I prefer counterparts of them in training for image generations also.
>
> **A2.**
> We appreciate the reviewer for the thoughtful comment. **We conducted additional experiments for Lipschitz continuity of OTM and UOTM on CIFAR-10, and added results in Appendix (Fig 11).**
> We measured the norm of the potential gradient $\\| \nabla_{y} v_{\phi}(y)\\|, \\| \nabla_{\hat{y}} v_{\phi}(\hat{y})\\|$ at a random real data $y$ and a randomly generated data $\hat{y}$. **The result shows a similar trend as in Fig 6.** The gradient norm of OTM potential is much larger than the gradient norm of UOTM potential throughout training on CIFAR-10.
>
>
> $ $
>
> ---
> **W3.**
> UOTM-SD look working very good in CIFAR-10, but it is not evident that is also good for other domains.
>
> **Q2.**
> The author achieved best performance of UOTM-SD on CIFAR-10, it would be good. But how about different image data? Does the scheduling strategy also improve stability using other image data?
>
> **A3.**
> We appreciate the reviewer for the thoughtful comment. We evaluated our UOTM-SD model on the high-resolution dataset, **CelebA-HQ ($256 \times 256$)**. Due to limitations in time and resources, we conducted one experiment for each model with NCSN++ backbone: UOTM-SD with Linear scheduling ($g_1=g_2=SP$), UOTM ($g_1=g_2=SP$), and OTM. We selected these three models as they represent competitive options among OT-based adversarial networks on CIFAR-10 (Table 3). The FID scores below demonstrate our UOTM-SD model outperforms the other two OT map models.
>
> |Model|FID ($\downarrow$)|
> |:---|:---|
> |OTM|13.56|
> |UOTM (SP)|9.78|
> |UOTM-SD (Linear)|**8.19**|
>
> The training hyperparameters are as follows: the cost intensity $\tau=0.00001$, R1 regularization intensity $\lambda=5$, Scheduling intensity $(\alpha_{min}, \alpha_{max})=(1/5, 5)$.

---

> > ### Author Response · Authors · 2023-11-22
> >
> > We thank the reviewer for the efforts in reviewing our paper. We would appreciate it if the reviewer let us know whether our response was helpful in addressing the reviewer's concerns. If there are additional concerns or questions, please let us know.

---

> > ### Comment · Reviewer_1bYC · 2023-11-22
> >
> > Thanks a lot for your response.
> >
> > I understand the answers:
> > - **A1.** On $g_{1,2}$ and $\Psi_{1,2}^*$, I understand my misunderstandings. Thank you for that, and I guess it would be better to comment on it in the main document.
> > - **A3.** I understand UOTM-SD works other domains, at least CelebA-HQ.
> >
> > But I cannot figure out
> > - **A2.** Fig 11 in the main pdf file seems to be generated images, not showing Lipschitz continuity.
> >
> > So please clarify it if possible.

---

> > > ### Author Response · Authors · 2023-11-22
> > >
> > > We appreciate the reviewer for the comment.
> > >
> > > **A1.**
> > > Thank you for the advice. We revised our manuscript to clarify that $g_1, g_2$ correspond to $\Psi_{1,2}^{*}$ as follows on Page 3 of Sec 3.1:
> > >
> > > > Note that $g_1, g_2$ correspond to $\Psi_{1,2}^{*}$ in Eq 8.
> > >
> > > **A2.**
> > > We are sorry for the confusion. The additional experiments for Lipschitz continuity are included in Fig 11 of the **revised version of our manuscript**.

---

> > > > ### Comment · Reviewer_1bYC · 2023-11-22
> > > >
> > > > > A1.
> > > >
> > > > Thank you for your consideration. I will keep my score, but raise Soundness.
> > > >
> > > > > A2. We are sorry for the confusion. The additional experiments for Lipschitz continuity are included in Fig 11 of the revised version of our manuscript.
> > > >
> > > > I see. Thank you for your quick reply.

---

> > > > > ### Author Response · Authors · 2023-11-22
> > > > >
> > > > > Thank you for the response. We appreciate the reviewer for reviewing our paper.

---

### Official Review · Reviewer_7jfY · 2023-10-30

**Soundness:** 3 good
**Presentation:** 3 good
**Contribution:** 3 good
**Rating:** 6
**Confidence:** 4

**Summary:**

In this paper, the authors propose novel theoretical results on varies aspects of the recently proposed Unbalanced Optimal Transport Model (UOTM), which is an optimal transport (OT) based generative model. In particular, in section 3, the authors provide an insight of how different choice of g_1 and g_2 and cost function could help stabilize training. In theorem 3.1, the authors prove the existence and uniqueness of the UOTM model. Moreover in section 4, the authors propose a novel alpha-scheduling method to stabilize training as well as mitigating the mode collapse/mixture problem. Theorem 4.1 shows that under this schema, the solutions of the UOT problems converge to the OT solution when alpha goes to infinity. This provides a new approach of solving the OT problem in the context of generative models.

**Strengths:**

The originality of the paper mainly comes from theorems 3.1 and 4.1, which consolidate the recently proposed UOTM method. These theorems pave the way of the proposed method that addresses the tau-sensitivity problem. Clarity of the paper looks good to me overall, but there are some places I'm confused about.

**Weaknesses:**

The experiment results look promising as well. It would be great if the proposed method is applied on higher solution image dataset to showcase the image generation quality.

**Questions:**

1. For WGAN explanation in section 3.1, the authors categorize this case as c = 0. I don't think this is the case because the in the original paper, this cost is the L1 Euclidean distance, i.e. c(x, y) = |x-y|. Also, it seems the Lipschitz constraint is missing in this case. It would be great if this part is further clarified.
2. Same comment for the italic sentence after Eq. 5.
3. More of a suggestion: in the experiment part (E.g. Fig. 1, Fig. 5 etc.), it would be great if the authors also include non-DNN based OT maps like the one proposed in An et. al 2019, as their solution is unique and can be found by a convex optimization.

---

> ### Author Response · Authors · 2023-11-17
> **Response to Reviewer 7jfY**
>
> We are deeply grateful to the reviewer for reading our paper and offering thoughtful feedback. We highlighted the corresponding revisions in the manuscript in Brown.
>
> $ $
>
> ---
> **W1.**
> The experiment results look promising as well. It would be great if the proposed method is applied on higher solution image dataset to showcase the image generation quality.
>
> **A1.**
> We thank the reviewer for the thoughtful comment. We evaluated our UOTM-SD model on the high-resolution dataset, **CelebA-HQ ($256 \times 256$)**. Due to limitations in time and resources, we conducted one experiment for each model with NCSN++ backbone: UOTM-SD with Linear scheduling ($g_1=g_2=SP$), UOTM ($g_1=g_2=SP$), and OTM. We selected these three models as they represent competitive options among OT-based adversarial networks on CIFAR-10 (Table 3). The FID scores below demonstrate our UOTM-SD model outperforms the other two OT map models.
>
>
> |Model|FID ($\downarrow$)|
> |:---|:---|
> |OTM|13.56|
> |UOTM (SP)|9.78|
> |UOTM-SD (Linear)|**8.19**|
>
> The training hyperparameters are as follows: the cost intensity $\tau=0.00001$, R1 regularization intensity $\lambda=5$, Scheduling intensity $(\alpha_{min}, \alpha_{max})=(1/5, 5)$.
>
> $ $
>
> ---
> **Q1.**
> For WGAN explanation in section 3.1, the authors categorize this case as $c = 0$. I don't think this is the case because the in the original paper, this cost is the L1 Euclidean distance, i.e. $c(x, y) = |x-y|$. Also, it seems the Lipschitz constraint is missing in this case. It would be great if this part is further clarified.
>
> **Q2.**
> Same comment for the italic sentence after Eq. 5.
>
> **A2.**
> We thank the reviewer for the careful comment. **We would like to clarify that the WGAN explanation in Sec 3.1 is presented in terms of the unified training algorithm (Alg. 1).** If we set $c=0$ in Line 5 and 10 in Alg. 1, the learning objective $\mathcal{L}$ becomes Eq. 4. Note that the L1 Euclidean distance $c(x, y) = \\|x-y\\|$ in WGAN is different from our cost function in Alg. 1. This L1 Euclidean distance is utilized in the definition of the Wasserstein-1 distance. Eq. 4 is derived from the Kantorovich-Rubinstein duality for the Wasserstein-1 distance.
>
> Moreover, the **Lipschitz constraint** is implemented by weight clipping in the original WGAN paper and by gradient penalty in WGAN-GP. We incorporated the implementation details regarding the Lipschitz constraint of WGAN on Page 4 as follows:
>
> > For the vanilla WGAN, we employed a weight clipping strategy for the potential network to impose the Lipschitz constraint, folllowing (Arjovsky et al., 2017).
>
> Furthermore, we rephrased the italic sentence after Eq. 5 to enhance clarity as follows:
>
> > Note that, if we set $c=0$ and introduce a 1-Lipschitz constraint on $v_{\phi}$, this objective has the same form as WGAN (Eq. 4).
>
> $ $
>
> ---
> **Q3.**
> More of a suggestion: in the experiment part (E.g. Fig. 1, Fig. 5 etc.), it would be great if the authors also include non-DNN based OT maps like the one proposed in An et. al 2019, as their solution is unique and can be found by a convex optimization.
>
> **A3.**
> Thank you for the valuable comment. We agree with the reviewer that comparing the GT transport map with the OT map models is a meaningful approach to evaluate the quality of transport map from the OT maps models. Due to page constraints, **we included this result in Appendix D.1.** In the Toy datasets, **both UOTM and UOTM-SD succeeded in learning the optimal transport map**, with minor differences to the GT transport map. This GT transport map is discovered through convex optimization [1].
>
>
> [1] Flamary, Rémi, et al. "Pot: Python optimal transport." The Journal of Machine Learning Research 22.1 (2021): 3571-3578.

---

> > ### Author Response · Authors · 2023-11-22
> >
> > We thank the reviewer for the efforts in reviewing our paper. We would appreciate it if the reviewer let us know whether our response was helpful in addressing the reviewer's concerns. If there are additional concerns or questions, please let us know.

---

### Official Review · Reviewer_12Gz · 2023-10-31

**Soundness:** 3 good
**Presentation:** 3 good
**Contribution:** 2 fair
**Rating:** 6
**Confidence:** 3

**Summary:**

This paper frames together two similar adversarial generative models: GANs involving a generator that learns to minimize an OT distance, and models whose objective is to directly learn the OT map from a prior distribution to the data distribution. The authors evaluate the stability and mode collapse of these models and conclude on the advantage of OT map models. In this category, one (based on unbalanced OT) is more performant than the other (based on standard OT), but less robust to hyperparameter choices. To alleviate this issue, the authors propose an interpolation strategy between both models to be scheduled during training.

**Strengths:**

By studying the considered adversarial models through a unified framework, this paper provides **interesting comparative insights** on their experimental performance. These insights might be valuable for future research in this area, highlighting the value of OT map models. These insights are **well illustrated** thanks to toy experiments, and the paper is overall **clear and easy to read**. The resulting proposed model refinements provide **improvements in generative performance and robustness to hyperparameter choices**, which is a substantial contribution in this domain.

**Weaknesses:**

The paper suffers from three main weaknesses that, together, make me believe that it remains under the acceptance threshold. I look forward to discussing with the authors and other reviewers on this topic.

### Significance of the Proposed Framework

As it is described in Algorithm 1, the proposed framework is a useful framework for exposition and experimental design, but the significance of this contribution is limited.
- It is a straightforward generalization of the already established lookalike adversarial objectives of Equations 4 and 5. This kind of framework is common in the GAN literature, e.g. in Nagarajan et al. (2017).
- The unified algorithm is not by itself the source of novel insights, or novel models. The proposed UOTM-SD does not necessitate this unified algorithm as it only interpolates between two OT map models using Equation (12).

Nagarajan et al. Gradient descent GAN optimization is locally stable. NIPS 2017.

### Weak Experiments

The experiments only weakly support the claims of the paper.
- The convergence and mode collapse properties are mainly studied in Section 3.2 on a toy dataset. Yet, the difference between low and high dimensions can be large when dealing with neural networks. Considering a higher-dimensional structured dataset could strengthen the experimental conclusions. Additionally, I would also suggest including another low-dimensional dataset to avoid any bias linked to having a data distribution evenly distributed around the prior distribution.
- Still on Section 3.2, the experiments lack a vanilla GAN baseline, especially to conclude on advantage of the SP function. Similarly, experiments of Figures 5 and 6 miss the WGAN-GP baseline.
- As a standalone model and with the available information, the comparative advantage of UOTM-SD is not significant enough. The gain in FID is minor and would thus require confidence intervals to be validated. The FID being computed on the training set, there is also a risk of overfitting to be taken into account. Furthermore, other datasets might be considered to test the robustness of the methods to other modalities and data dimensions.

### Possible Bias against OT Loss Models (GANs)

Generators in GANs do not require their latent space to be of the same dimension as their output. Yet, it seems to be the case for OT map models, given that they learn a transport map between two distributions living in the same space. I would suggest the authors to explicitly explain how this affects their experiments and their results. Are the provided comparisons fair between the two types of models, in terms of dimensionality and neural network architectures? Both operation modes seem hard to articulate with each other, as Algorithm 1 features the sampling of both a random variable from the same space as the data and another latent variable to accommodate the two types of models.

Moreover, the authors should further the comment why the Lipschitzness result of Theorem 3.1 is an advantage over GANs. WGAN(-GP) also requires Lipschitz solutions, and the Lipschitzness constraint is even applied to other GAN models nowadays.

### Remarks on the Form

- The references of Fan et al. and Rout et al. miss a year.
- Some notations are not defined, like $\Pi(\mu, \nu)$ and $D_{\psi_i}$ in Section 2.
- Differentials $d$ in integrals should be upright for better readability.
- Abbreviations of "Equation" should end with a point: "Eq.".
- Equations 4 and 5 are not learning objectives but optima of learning objective. The correct way to present them is in Algorithm 1.
- The color scheme of Figure 5 should be adjusted for a better readability in grayscale.
- Some space should be added between images and captions in Figure 4.

### Post-Rebuttal

I would like to thank the authors for our discussion. **Their responses addressed a number of my concerns**, especially by strengthening the experiments and ensuring the soundness of their conclusions. Two main weaknesses remain in my opinion: the lack of strong connection between the two main contributions of the paper (Sections 3 and 4), and the Lipschitzness results which are not conclusive enough. Still, the paper is interesting and I think it might inspire new research on the topic. Hence, **I raise my "soundness" score from 2 to 3 and my main recommendation from "marginally below the acceptance threshold" to "marginally above the acceptance threshold"**.

**Questions:**

Cf. the *Weaknesses* part of the review for questions related to paper improvements.

---

> ### Author Response · Authors · 2023-11-17
> **Response to Reviewer 12Gz (1/4)**
>
> We appreciate the reviewer for spending time reading our manuscript carefully and providing thoughtful feedback. We hope our replies to be helpful in addressing the reviewer's concerns. We highlighted the corresponding revisions in the manuscript in Blue.
>
> $ $
>
> ## Significance of the Proposed Framework
> ---
> **Q1.** It is a straightforward generalization of the already established lookalike adversarial objectives of Equations 4 and 5. This kind of framework is common in the GAN literature, e.g. in Nagarajan et al. (2017).
>
> **A1.**
> **We would like to emphasize that the OT map models are derived from a fundamentally different approach when compared to previous GAN models.**
>
> - In GAN models (OT Loss), we consider the OT problem between the generated distribution ($\mu$) and the data distribution ($\nu$). The OT problem is employed to measure the distance between $\mu$ and $\nu$. **The previous works suggesting the generalization of GAN models, such as Nagarajan et al. (2017), only cover these OT Loss GANs.**
> - In contrast, OT map models consider the OT problem between the Gaussian distribution ($\mu$) and the data distribution ($\nu$). The OT map itself serves as a generative model.
>
> Our key observation is that, although these models are derived from different approaches, these models can be integrated into a unified adversarial framework. **To the best of our knowledge, our work is the first attempt to propose a unified framework for OT Loss GANs and OT map models.** This framework provides a better understanding of **why recent OT map models exhibit significantly different dynamics compared to OT Loss GANs**.
>
>
> $ $
>
> ---
> **Q2.** The unified algorithm is not by itself the source of novel insights, or novel models. The proposed UOTM-SD does not necessitate this unified algorithm as it only interpolates between two OT map models using Equation (12).
>
> **A2.**
> Because we focused on the contribution of the unified framework in A1, **we would like to emphasize the contribution of analysis stemming from this framework and its connection to UOTM-SD.**
>
> This unified framework allows us to conduct ablation studies on two building blocks of this framework: (1) Strictly convex $g_1$, $g_2$ and (2) Cost function $c(\cdot, \cdot)$. This comparative analysis uncovers the role of each component in generative modeling.
>
> - The strictly convex $g_1$, $g_2$ helps stabilize the training process over setting $g_1 = g_2 = Id$.
> - The cost function helps mitigate the mode collapse problem in the adversarial training framework.
>
> However, simultaneously exploiting these two advantages (UOTM) reveals some limitations.
> - The generative performance is sensitive to the hyperparameter $\tau$.
> - The UOT problem (Eq. 6) inherently incurs distribution errors.
>
> In this respect, as discussed in the Motivation paragraph of Sec 4, we introduced UOTM-SD to address these limitations. **Our UOTM-SD minimizes distribution errors by designing a converging sequence of the UOT map towards the OT map through $\alpha$-scheduling (Theorem 4.1.). Moreover, for each $\alpha$, UOTM-SD can leverage the advantages of two building blocks, as UOTM.**

---

> ### Author Response · Authors · 2023-11-17
> **Response to Reviewer 12Gz (2/4)**
>
> ## Weak Experiments
> ---
> **Q3.** The convergence and mode collapse properties are mainly studied in Section 3.2 on a toy dataset. Yet, the difference between low and high dimensions can be large when dealing with neural networks. Considering a higher-dimensional structured dataset could strengthen the experimental conclusions.
>
> **A3.**
> We appreciate the reviewer for the thoughtful advice. **In Table 2 and Fig 4, we presented the convergence and mode collapse analyses for the image dataset (CIFAR-10).** In Table 2, we measured FID scores for the convergence analysis. As discussed in Page 5, UOTM w/o cost and UOTM outperform their algorithmic counterparts concerning $g_1$, $g_2$, i.e., WGAN and OTM, respectively. Moreover, we presented the generated samples for each $\tau$ in Fig 4 for mode collapse analysis. As dicussed in Page 6, when $\tau$ is too small, the model exhibits a mode collapse problem, generating similar samples repeatedly.
>
>
> $ $
>
> ---
> **Q4.** I would also suggest including another low-dimensional dataset to avoid any bias linked to having a data distribution evenly distributed around the prior distribution.
>
> **A4.**
> **We conducted additional experiments on two Toy datasets: Uneven Gaussian Mixture and Moon-to-Spiral.** The comprehensive comparisons of generators, including WGAN-GP, UOTM-SD, and GT transport map from a convex optimization [1], are provided in Fig 8, 9, and 10 of Appendix D.1. These additional experiments yield consistent results with our analysis in Sec. 3.2.
>
>
> $ $
>
> ---
> **Q5** Still on Section 3.2, the experiments lack a vanilla GAN baseline, especially to conclude on advantage of the SP function.
>
> **A5.**
> We would like to emphasize that **the goal of this work is to investigate OT-based adversarial networks, specifically derived from OT Loss GANs and OT map models.** Consequently, while the vanilla GAN could be accommodated within our unified framework (Sec 3.1), we considered the vanilla GAN to be beyond the scope of our current analysis. However, as we discussed in Sec 5, we agree with the reviewer that a comparative analysis with the vanilla GAN could be an interesting future research. This additional analysis could be helpful in investigating the effect of $g_3$. (Note that, except for the vanilla GAN, all OT-based adversarial networks in our analysis set $g_3=Id$).
>
> Moreover, **concerning  $g_1$ and $g_2$, we discussed both advantageous aspects and drawbacks of setting $g_1, g_2$ as the SP function compared to the Identity function, throughout Sec 3.2.** Specifically, employing SP for $g_1, g_2$ contributes to stabilizing the training process through the adaptive gradient updates (Sec 3.2.1) and the Lipschitzness of potential (Sec. 3.2.3). However, the drawback is that the UOTM model (using $g_1 = g_2 = SP$ and the cost function) introduces inherent distribution errors (Eq 11). In this regard, we proposed UOTM-SD to address the drawback while leveraging the advantages.
>
>
> $ $
>
> ---
> **Q6.** Experiments of Figures 5 and 6 miss the WGAN-GP baseline.
>
> **A6.**
> We appreciate the reviewer for the comment. We added WGAN-GP results for Fig 5 in Fig 8 of Appendix D.1, and revised the manuscript accordingly. **In Fig 8, the result of WGAN-GP is consistent with our analysis, that the cost function helps mitigate the mode collapse/mixture problem in OT-based adversarial networks.** First, WGAN-GP exhibits the mode mixture problem. Moreover, the generator $T_{\theta}$ of WGAN-GP randomly matches the input noise to the generated sample. This random matching is visually evident from the intricate intersections of connecting lines around the input noise samples (depicted in Green).
>
> Furthermore, we included WGAN-GP results for Fig 6 in Fig 12 of Appendix D.3. As anticipated, Fig 12 shows that the Lipschitz constant of WGAN-GP potential is bounded. **The Gradient Penalty regularizer in WGAN-GP directly controls the Lipschitz constant of $v_{\phi}$ to impose the $1$-Lipschitz constraint on potentials in WGAN (Eq. 4).** Hence, if we set penalty hyperparameter $\lambda$ large, the Lipschitz constant remains bounded. However, this is a consequence of the Gradient Penalty and not an inherent property of the learning objective itself. **We would like to emphasize that this result is completely different from UOTM in Fig. 6.** Theorem 3.1 states that the Lipschitz constant of proper potential $v_{\phi}$ is equi-bounded, even when **we conduct unconstrained optimization over $v_{\phi}$.**

---

> ### Author Response · Authors · 2023-11-17
> **Response to Reviewer 12Gz (3/4)**
>
> ---
> **Q7.** The gain in FID is minor and would thus require confidence intervals to be validated. The FID being computed on the training set, there is also a risk of overfitting to be taken into account. Furthermore, other datasets might be considered to test the robustness of the methods to other modalities and data dimensions.
>
> **A7.** We thank the reviewer for the comment. We performed two additional experiments for each UOTM-SD model in Table 3 on CIFAR-10. Our UOTM-SD model demonstrates noticeably improved results comapred to UOTM model, considering the standard deviation of FID scores. The mean and standard deviation of FID scores (mean $\pm$ standard deviation) are as follows:
>
> |Model|FID ($\downarrow$)|
> |:---|:---|
> |UOTM|$2.99 \pm 0.07$|
> |UOTM-SD (Cosine)|$2.57 \pm 0.09$|
> |UOTM-SD (Linear)|$2.50 \pm 0.11$|
> |UOTM-SD (Step)|$2.77 \pm 0.4$|
>
> Moreover, we evaluated our UOTM-SD model on the high-resolution dataset, **CelebA-HQ ($256 \times 256$)**. Due to limitations in time and resources, we conducted one experiment for each model with NCSN++ backbone: UOTM-SD with Linear scheduling ($g_1=g_2=SP$), UOTM ($g_1=g_2=SP$), and OTM. We selected these three models as they represent competitive options among OT-based adversarial networks on CIFAR-10 (Table 3). The FID scores below demonstrate our UOTM-SD model outperforms the other two OT map models.
>
> |Model|FID ($\downarrow$)|
> |:---|:---|
> |OTM|13.56|
> |UOTM (SP)|9.78|
> |UOTM-SD (Linear)|**8.19**|
>
> The training hyperparameters are as follows: the cost intensity $\tau=0.00001$, R1 regularization intensity $\lambda=5$, Scheduling intensity $(\alpha_{min}, \alpha_{max})=(1/5, 5)$.
>
>
> $ $
>
> ## Possible Bias against OT Loss Models (GANs)
> ---
> **Q8.** Generators in GANs do not require their latent space to be of the same dimension as their output. Yet, it seems to be the case for OT map models, given that they learn a transport map between two distributions living in the same space. I would suggest the authors to explicitly explain how this affects their experiments and their results. Are the provided comparisons fair between the two types of models, in terms of dimensionality and neural network architectures? Both operation modes seem hard to articulate with each other, as Algorithm 1 features the sampling of both a random variable from the same space as the data and another latent variable to accommodate the two types of models.
>
> **A8.** We sincerely appreciate the reviewer for the thoughtful comment regarding the missing clarifications and details in our manuscript. We thoroughly revised the main part and appendix of our manuscript to incorporate these previously missing descriptions.
>
> - **Description of auxiliary variable $z\sim \mathcal{N}(0,I)$:**
> This random variable $z$ in Algorithm 1 does not represent the input Gaussian Noise in GAN models. Instead, this auxiliary noise is provided to the generator $T_{\theta}$ in the OT map models, such as UOTM [2], to enhance generative performance from the practical perspective. We included a description of this auxiliary variable $z$ in Sec 3.1 as follows:
>
> > Also, the Gaussian noise $z$ represents the auxiliary variable and is different from the input prior noise $x \sim \mu$. This auxiliary variable $z$ is introduced to represent the stochastic transport map $T_{\theta}$ in the OT map models, such as UOTM.
>
> - **Description of latent space $\mathcal{X}$:**
> As the reviewer commented, the original formulation of OT map models (Eq 5 and 8) requires their latent space $\mathcal{X}$ to be of the same dimension as their output $\mathcal{Y}$. However, **these OT map models can also accommodate a smaller dimension for $\mathcal{X}$ by employing a simple practical scheme suggested in OTM [3].** We adopted this scheme for the DCGAN backbone for CIFAR-10 in Table 2, following the implementation of OTM. Therefore, **we believe that there are no constraints on the latent dimension and, consequently, no restrictions on the neural network architecture.** We clarified these implementation details in Appendix B and added descriptions on Page 7 as follows:
>
> > [Page 7] Note that the DCGAN backbone employs the deterministic upsampling strategy from OTM [3] to accommodate an input latent space with a smaller dimension than the data space. (See Appendix B for details).
>
> (Continued on Response (4/4))

---

> ### Author Response · Authors · 2023-11-17
> **Response to Reviewer 12Gz (4/4)**
>
> > [Appendix B] In the DCGAN backbone, we adopt a simple practical scheme suggested in OTM [3] for accommodating a smaller dimension for the input latent space $X$. This practical scheme involves introducing a deterministic bicubic upsampling $Q$ from $\mathcal{X}$ to $\mathcal{Y}$. Then, we consider the OT map between $Q_{\sharp}\mu$ and $\nu$. In practice, we sample $x$ in Algorithm 1 from a 192-dimensional standard Gaussian distribution. Then, $x$ is directly used as an input for the DCGAN generator $T_\theta$. The random variable $z$ is not employed in the DCGAN implementation. Meanwhile, $Q(x)$ is obtained by reshaping $x$ into a $3\times 8\times 8$ dimensional tensor, and then bicubically upsampling it to match the shape of the image. The generator loss is defined as $c\left(Q(x), T_\theta (x) \right) - v_\phi \left( T_\theta (x) \right)$.
>
> > [Appendix B] Specifically, we set $\mathcal{X}=\mathcal{Y}$ and use $c(x,y) = \tau \lVert x-y \rVert^2$ without introducing upsampling $Q$.  Here, the auxiliary variable $z$ is employed. We sample $z$ from a 256-dimensional Gaussian distribution and put it as an additional stochastic input to the generator. The input prior sample $x$ is fed into the NCSN++ network like UNet input. The auxiliary $z$ passes through embedding layers and is incorporated into the intermediate feature maps of the NCSN++ through an attention module.
>
>
> $ $
>
> ---
> **Q9** Moreover, the authors should further the comment why the Lipschitzness result of Theorem 3.1 is an advantage over GANs. WGAN(-GP) also requires Lipschitz solutions, and the Lipschitzness constraint is even applied to other GAN models nowadays.
>
> **A9.**
> **We would like to distinguish between constrained optimization (WGAN(-GP)) and unconstrained optimization (Theorem 3.1).** For example, consider two optimization problems: for some $u \in \mathbb{R}^{d}$,
> $$
>     \text{maximize}\_{v \in \mathbb{R}^{d}, \\| v \\|\_2 = 1}   \langle u, v \rangle  \qquad
>     \text{minimize}\_{v \in \mathbb{R}^{d}} \\| u - v \\|\_{2}^{2}.
> $$
> The constrained optimization problem on the left is meaningful only when we impose the constraint $\\| v \\|\_2 = 1$. Under the constraint, the maximum value is $\\| u \\|\_2$. Without the constraint, $\langle u, v \rangle$ simply diverges to $\infty$ as we increase $\\| v \\|\_2$.
> On the contrary, the optimization problem on the right does not impose any constraint on $v$. Nevertheless, minimizing the objective function $\\| u-v \\|\_{2}^{2}$ naturally provides some regularity on $v$: if $\\| u-v\_{i} \\|\_{2}^{2} < r^{2}$ for some iterate $v\_{i}$, $v\_{i}$ situated within a ball near $u$. A similar phenomenon occurs between WGAN(-GP) and UOTM.
>
> **The learning objective of WGAN(-GP) conducts optimization over the 1-Lipschitz potential.** The Gradient Penalty regularizer is introduced to constrain the potential network to satisfy the 1-Lipschitz constraint. As discussed in Appendix C, several studies showed that such Lipschitz constraints may lead to mode collapse/mixture problems [4, 5].
>
> On the contrary, **Theorem 3.1 states that, under unconstrained optimization, the potential networks $v_{\phi}$ with only minor conditions satisfy the equi-Lipschitzness**. Therefore, Theorem 3.1 suggests that the potential networks $v_{\phi}$ would consistently represent properly regular behavior throughout training, avoiding issues such as exploding gradients. This regularity of potential network in the UOTM model is verified through Fig 6. Following the reviewer's advice, we revised our manuscript on Page 7 as follows:
>
>
> > Note that Theorem 3.1 is fundamentally different from the 1-Lipschitz constraint of WGAN (Eq. 4). WGAN involves constrained optimization over a 1-Lipschitz potential. In contrast, Theorem 3.1 states that, under unconstrained optimization, the potential networks $v_{\phi}$ with only minor conditions satisfy equi-Lipschitzness.
>
> $ $
>
> ## Remarks on the Form
> ---
> Thank you for the valuable advice regarding the presentation of our work. Following the advice, we revised our manuscript.
>
> $ $
>
> **References**
> [1] Flamary, Rémi, et al. "Pot: Python optimal transport." The Journal of Machine Learning Research 22.1 (2021): 3571-3578.
> [2] Choi, Jaemoo, Jaewoong Choi, and Myungjoo Kang. "Generative Modeling through the Semi-dual Formulation of Unbalanced Optimal Transport." NeurIPS 2023.
> [3] Rout, Litu, Alexander Korotin, and Evgeny Burnaev. "Generative modeling with optimal transport maps." ICLR 2022.
> [4] Nagarajan, Vaishnavh, and J. Zico Kolter. "Gradient descent GAN optimization is locally stable." NeurIPS 2017.
> [5] Khayatkhoei, Mahyar, Maneesh K. Singh, and Ahmed Elgammal. "Disconnected manifold learning for generative adversarial networks." NeurIPS 2018.

---

> ### Comment · Reviewer_12Gz · 2023-11-20
> **Discussion on the New Revision**
>
> I would like to thank the authors for their detailed answer and additional experiments. Let me fuel the discussion on each point below. Depending on the extent of our discussion, I might wait for the end of the discussion period and the reviewer discussion phase to revise my recommendation.
>
> ## Significance of the Proposed Framework
>
> I understand that this paper is the first to introduce a framework encompassing both OT loss and OT map models. However, I meant that, since both types of models are already very similar in their max-min formulation from Eq. 4 and 8, the significance of this contribution is limited. I was taking Nagarajan et al. (2017) as an example among (many) papers joining multiple GAN models in a single framework, where the framework is not the core contribution but rather a technical tool.
>
> In the case of this submission, I would rather see the introduced framework as a technical tool enabling the following ablation studies. This makes the conclusions of the ablation studies the main contribution, instead of the framework itself.
>
> Furthermore, I still think that the proposed UOTM-SD is not directly related to this framework. Every derivation yielding to this model only deals with OT map models, and does not necessitate the more global framework from the previous section.
>
>
> ## Weak Experiments
>
> ### Q3 (Convergence and Mode Collapse Analysis)
>
> My appreciation is that the described results on high-dimensional data are not sufficiently grounded in the ablation study of Section 3.2. Table 2 does show results on CIFAR-10, but only using the FID which is not granular enough to study e.g. mode collapse. The authors could consider using precision/recall-type metrics (Naeem et al., 2020). Furthermore, samples in Fig. 4 only deal with UOTM and not the ablation study.
>
> Naeem et al. Reliable Fidelity and Diversity Metrics for Generative Models. ICML 2020.
>
> ### Q6 (Inclusion of WGAN-GP)
>
> For comments regarding the fundamental difference between WGAN-GP and UOTM, please refer to the next section. Regarding the experimental results, it seems to me that the induced Lipschitzness of WGAN-GP, similar to the one of UOTM, clashes with the claim that it explains the stable training of models (Section 3.2.3). Given the performance difference between both models, the Lipschitzness of UOTM surely does not explain its advantage.
>
> ### Q5 (Comparison to Vanilla GAN)
>
> I remain convinced by the interest of a comparison to vanilla GAN to better analyze the role of the SP functions, but I also understand the scope chosen by the authors. To clarify this, I would suggest the authors to remove the references to vanilla GAN in the paper, and entirely keep the focus on OT-based approaches.
>
> ### Q4 & Q7 (Additional Datasets)
>
> I would like to thank the authors for the additional datasets which strengthen both the ablation study and the significance of the proposed UOTM-SD.
>
>
> ## Possible Bias against OT Loss Models (GANs)
>
> ### Q8 (Input and Output Dimensionality)
>
> I would like to thank the authors for the clarifications, which seem to address my concern. Could they confirm then that all experiments in Section 3 are done with equivalent latent space dimensions and architectures?
>
> ### Q9 (Lipschitzness)
>
> I understand the difference between WGAN(-GP) and UOTM as the former corresponds to a constrained optimization problem, while the latter is based on an unconstrained one. Nevertheless, the implementations of WGAN-type models are quite different from their theoretical optimization problem, as the Lipschitzness constraint is enforced in various ways.
>
> I would argue that WGAN-GP departs from this constrained optimization problem and is much closer to UOTM, which, in some sense, is also based on a regularization (the cost in the objective), than the original version of WGAN.
>
> This, together with my comments on the new experimental results of Figure 6, convinces me that the results fail to show that the Lipschitzness property is the reason of the success of UOTM compared to WGAN-GP (but it does seem to be an interesting property w.r.t. OTM for instance).

---

> ### Author Response · Authors · 2023-11-21
> **Response to Reviewer 12Gz (1/2)**
>
> We appreciate the reviewer for reading our rebuttal and for asking further questions.
>
> $ $
>
> ## Significance of the Proposed Framework
>
> ---
> **Q1.**
> I understand that this paper is the first to introduce a framework encompassing both OT loss and OT map models. However, I meant that, since both types of models are already very similar in their max-min formulation from Eq. 4 and 8, the significance of this contribution is limited. I was taking Nagarajan et al. (2017) as an example among (many) papers joining multiple GAN models in a single framework, where the framework is not the core contribution but rather a technical tool.
> In the case of this submission, I would rather see the introduced framework as a technical tool enabling the following ablation studies. This makes the conclusions of the ablation studies the main contribution, instead of the framework itself.
>
> **A1.** We thank the reviewer for acknowledging that this paper is the first to introduce a framework encompassing both OT loss and OT map models.
> Also, **we agree with the reviewers that the conclusions of ablation studies are the main contributions of this work.**
> The motivating question of this work was as follows:
>
> The OT loss and OT map models arrive at similar adversarial learning objectives. However, the OT map models exhibit superior performance compared to previous OT loss models. Then, what components contribute to the difference between these two models?
>
> **To address this question, we established our framework and studied each component to obtain a better understanding of their individual roles.**
>
> $ $
>
> ---
> **Q2.**
> Furthermore, I still think that the proposed UOTM-SD is not directly related to this framework. Every derivation yielding to this model only deals with OT map models, and does not necessitate the more global framework from the previous section.
>
> **A2.**
> Thank you for the follow-up. As discussed in the above of Eq 13, **UOTM-SD can be incorporated into our unified framework by scheduling $g$, i.e. $g^\alpha_i (x)= \alpha g_i (x/\alpha)$**.
> Moreover, we would like to emphasize that **we proposed an improvement over the best-performing OT-based model identified in our analysis.** UOTM model is selected not because it is the OT map model, but because UOTM exploits both beneficial components identified in our ablation studies:
> (1) Strictly convex $g_1, g_2$ and (2) Cost function $c(x,y)=\tau \\|x-y\\|_{2}^{2}$.
>
> However, employing both beneficial components simultaneously reveals an inherent limitation: the UOT problem induces distribution errors. In this respect, **our ablation studies elucidated why directly minimizing distribution errors by decreasing $\tau$ (Eq 11) leads to worse performance for UOTM.** When $\tau$ is too small, the cost function fails to provide sufficient mode coverage effect for UOTM. In this regard, we proposed UOTM-SD, which minimizes distribution errors by scheduling the divergence terms.
>
> $ $
>
> ## Weak Experiments
> ---
> **Q2. (Convergence and Mode Collapse Analysis)**
> My appreciation is that the described results on high-dimensional data are not sufficiently grounded in the ablation study of Section 3.2. Table 2 does show results on CIFAR-10, but only using the FID which is not granular enough to study e.g. mode collapse. The authors could consider using precision/recall-type metrics (Naeem et al., 2020). Furthermore, samples in Fig. 4 only deal with UOTM and not the ablation study.
>
> **A3.**
> We are grateful for the insightful feedback. Following the reviewer's feedback, **we evaluated precision/recall metrics on CIFAR-10 for five OT-based adversarial networks.** The results are consistent with our analysis on the Toy datasets: (Due to time constraints during the discussion phase, we first presented these precision/recall metrics through the comment. We will incorporate these results into Table 2.)
>
>
> |Model      |Precision  |Recall|
> |:---|:---|:---|
> |WGAN|0.45|0.02|
> |WGAN-GP|0.71|0.55|
> |UOTM w/o cost|0.80|0.13|
> |OTM|0.71|0.49|
> |UOTM (SP)|0.78|0.62|
>
> The recall metric assesses the mode coverage for each model. In this regard, **the introduction of the cost function improves the recall metric for each model**: from WGAN (0.02) to OTM (0.49) and from UOTM w/o cost (0.13) to UOTM (0.62). On the other hand, the precision metric evaluates the faithfulness of generated images for each model. Interestingly, UOTM w/o cost achieves the best precision score, but the recall metric is significantly lower than UOTM. This result shows that UOTM w/o cost exhibited the mode collapse problem.

---

> ### Author Response · Authors · 2023-11-21
> **Response to Reviewer 12Gz (2/2)**
>
> ---
> **Q6. (Inclusion of WGAN-GP)**
> For comments regarding the fundamental difference between WGAN-GP and UOTM, please refer to the next section. Regarding the experimental results, it seems to me that the induced Lipschitzness of WGAN-GP, similar to the one of UOTM, clashes with the claim that it explains the stable training of models (Section 3.2.3). Given the performance difference between both models, the Lipschitzness of UOTM surely does not explain its advantage.
>
> **A4.** Responded with Q9 at A9.
>
> $ $
>
> ---
> **Q5. (Comparison to Vanilla GAN)**
> I remain convinced by the interest of a comparison to vanilla GAN to better analyze the role of the SP functions, but I also understand the scope chosen by the authors. To clarify this, I would suggest the authors to remove the references to vanilla GAN in the paper, and entirely keep the focus on OT-based approaches.
>
> **A5.**
> Thank you for the advice. To avoid confusion, we removed the vanilla GAN from the unified framework in the revised manuscript.
>
> $ $
>
> ## Possible Bias against OT Loss Models (GANs)
> ---
> **Q8. (Input and Output Dimensionality)**
> I would like to thank the authors for the clarifications, which seem to address my concern. Could they confirm then that all experiments in Section 3 are done with equivalent latent space dimensions and architectures?
>
> **A6.**
> Yes, all experiments in Sec 3 were conducted with the same latent space dimensions and architectures across the OT-based approaches. We are encouraged that our response was helpful in addressing the reviewer's concerns.
>
> $ $
>
> ---
> **Q9. (Lipschitzness)**
> I understand the difference between WGAN(-GP) and UOTM as the former corresponds to a constrained optimization problem, while the latter is based on an unconstrained one. Nevertheless, the implementations of WGAN-type models are quite different from their theoretical optimization problem, as the Lipschitzness constraint is enforced in various ways.
> I would argue that WGAN-GP departs from this constrained optimization problem and is much closer to UOTM, which, in some sense, is also based on a regularization (the cost in the objective), than the original version of WGAN.
> This, together with my comments on the new experimental results of Figure 6, convinces me that the results fail to show that the Lipschitzness property is the reason of the success of UOTM compared to WGAN-GP (but it does seem to be an interesting property w.r.t. OTM for instance).
>
> $ $
>
> **A7.**
> We appreciate the reviewer for the follow-up questions. We would like to emphasize that Theorem 3.1 assumes the strictly convex $g_1, g_2$, and this assumption is not satisfied by OTM. Therefore, **we believe the benefit of Lipschitzness can be verified through a comparison between UOTM and OTM.** To avoid confusion, we revised Page 8 of our manuscript as follows:
>
> >This equi-Lipschitz continuity also explains the stable training of UOTM over OTM.
>
> **When we directly compare UOTM with WGAN-GP, several changes should be considered simultaneously,** such as (1) Cost function $c(x,y)$, (2) Strictly convex $g_1, g_2$, and (3) Gradient Penalty. While both models satisfy Lipschitzness during optimization, additional Cost function $c(x,y)$ and Strictly convex $g_1, g_2$ provide optimization advantage and mode mitigation effect to UOTM. We think that these components contribute to the superior performance of UOTM over WGAN-GP.
>
> Instead, **we consider the comparison between WGAN and WGAN-GP to be a meaningful example for testing the benefits of Lipschitzness as well.** The original WGAN introduced a weight clipping strategy to indirectly induce 1-Lipschitzness to the potential. However, as shown in Fig 6, the weight clipping strategy is not sufficient for imposing Lipschitzness on the potential. On the other hand, WGAN-GP succeeds in imposing Lipschitzness. Hence, WGAN-GP exhibits more stable convergence on the Toy dataset in Fig 1 and better performance on CIFAR-10 in Table 2, compared to WGAN.

---

> ### Comment · Reviewer_12Gz · 2023-11-23
> **Acknowledgement**
>
> I would like to thank the authors for their follow-up answer. I am looking forward to the discussion with the other reviewers to consolidate a final recommendation.

---

### Official Review · Reviewer_oAtf · 2023-11-05

**Soundness:** 3 good
**Presentation:** 3 good
**Contribution:** 2 fair
**Rating:** 5
**Confidence:** 3

**Summary:**

OT has been widely explored in generative modeling from diverse perspectives (e.g., OT loss and OT map). However, interpretation and understanding of the pros and cons of OT are underexplored.  In this paper, the authors are proposing a framework generalizing the existing OT-based generative models and additionally propose a scheduling method to mitigate the drawback of the cost function derived from OT. Both quantitative and qualitative analyses and experiments are reported.

**Strengths:**

- Background knowledge is contained well in the paper.
- Existing OT-based methods are analyzed well under the proposed generalizing framework.
- An additional scheduling method is proposed for mitigating a drawback of the cost function while boosting the benefit.

**Weaknesses:**

- It is not easy to understand without expertise in OT. It would be much easier to read if one or two lines of descriptions comparing each term and notation with those of the regular GAN setup were provided.
- The analysis is good, but the benefit of the proposed method (UOTM-SD) is not clearly shown.
- Experiments are limited to low-dimensional datasets which is not practical enough.
- Some parts are not clear which are described in the Questions below.

**Questions:**

1. Is the analysis valid in relatively higher-dimensional data (e.g., 128^2 or 256^2) such as  CelebA or FFHQ, CUB, or ImageNet?
2. (Alg. 1) What is $X$ in line 3? I would assume $Y$ is from real data and $z$ is from the prior known distribution, e.g., Gaussian. Similarly, $T_\theta$ is consistently taking $x$ as input throughout Equations in the paper while $x$ and $z$ are taken as input in the algorithm (line 4). For example, in Eq. 5, I can see that $x$ is a prior distribution $y$ is a real distribution and the OT term $c$ is applied in between the Gaussian and the generated samples, which I believe is different from the Algorithm.
3. (page 6, “Effect of Cost in Mode Collapse”) It is not straightforward how the cost function actually helps the mode coverage.
4. Why the results in Fig. 1 and Fig. 5 are different?
5. Performance of UOTM-SD in the toy dataset? and Qualitative comparison results in Cifar10 (UOTM-SD v.s. UOTM)?

---

> ### Author Response · Authors · 2023-11-17
> **Response to Reviewer oAtf (1/2)**
>
> We appreciate the reviewer for spending time reading our manuscript carefully and providing valuable feedback. We hope our replies to be helpful in addressing the reviewer's concerns. We highlighted the corresponding revisions in the manuscript in Red.
>
> $ $
>
> ---
> **W1.**
> It is not easy to understand without expertise in OT. It would be much easier to read if one or two lines of descriptions comparing each term and notation with those of the regular GAN setup were provided.
>
> **A1.**
> We appreciate the reviewer for the comment. We included additional explanations comparing the regular GAN and OT map models below Eq. 5 as follows:
>
>
> > Intuitively, $T_\theta$ and $v_\phi$ serve as the generator and the discriminator of a GAN. For convenience, we denote the optimization problem of Eq. 5 as an OT-based generative model (OTM). *Note that, if we set $c=0$ and introduce a 1-Lipschitz constraint on $v_{\phi}$, this objective has the same form as WGAN (Eq. 4).* In OT map models, the quadratic cost is usually employed, i.e., $c(x,y) = \tau \\|x-y\\|_{2}^{2}$.
>
> $ $
>
> ---
> **W2.**
> The analysis is good, but the benefit of the proposed method (UOTM-SD) is not clearly shown.
>
> **A2.**
> We appreciate the reviewer for acknowledging the novelty of our analysis. **We would like to emphasize that our UOTM-SD provides additional benefits in terms of generative performance (FID score in Table 3) and robustness to hyperparameter (Fig 7 and Table 5) over UOTM.**
>
> Our analysis demonstrated UOTM outperforms other OT-based adversarial networks by exploiting two key factors: (1) Stable training from the strictly convex $g_1, g_2$ and (2) Alleviation of mode collapse from the cost function $c(x, y)=\tau \\|x-y\\|_{2}^{2}$. **However, UOTM has some limitations.** The UOT problem incurs inherent distribution errors, which limit the generative performance of UOTM. Also, UOTM is sensitive to the cost intensity hyperparameter $\tau$.
>
>
> **In this respect, UOTM-SD addresses this inherent distribution error of UOTM stemming from the UOT problem (Theorem 4.1). Moreover, UOTM-SD is significantly more robust to $\tau$ than UOTM (Fig 7 and Table 5).** For example, when $\tau=2e-4$, UOTM yields an FID score of 15.19, whereas UOTM-SD achieves a substantially lower FID score of 3.60. Similarly, when $\tau=5e-3$, UOTM yields an FID score of 218.02, while UOTM-SD achieves a markedly improved FID score of 5.42.
>
>
> $ $
>
> ---
> **W3.**
> Experiments are limited to low-dimensional datasets which is not practical enough.
>
> **Q1.**
> Is the analysis valid in relatively higher-dimensional data (e.g., $128^2$ or $256^2$) such as CelebA or FFHQ, CUB, or ImageNet?
>
> **A3.**
> We thank the reviewer for the thoughtful advice. We evaluated our UOTM-SD model on the high-resolution dataset, **CelebA-HQ ($256 \times 256$)**. Due to limitations in time and resources, we conducted one experiment for each model with NCSN++ backbone: UOTM-SD with Linear scheduling ($g_1=g_2=SP$), UOTM ($g_1=g_2=SP$), and OTM. We selected these three models as they represent competitive options among OT-based adversarial networks on CIFAR-10 (Table 3). The FID scores below demonstrate our UOTM-SD model outperforms the other two OT map models.
>
>
> |Model|FID ($\downarrow$)|
> |:---|:---|
> |OTM|13.56|
> |UOTM (SP)|9.78|
> |UOTM-SD (Linear)|**8.19**|
>
> The training hyperparameters are as follows: the cost intensity $\tau=0.00001$, R1 regularization intensity $\lambda=5$, Scheduling intensity $(\alpha_{min}, \alpha_{max})=(1/5, 5)$.

---

> ### Author Response · Authors · 2023-11-17
> **Response to Reviewer oAtf (2/2)**
>
> ---
> **Q2.**
> (Alg. 1) What is $X$ in line 3? I would assume $Y$ is from real data and $z$ is from the prior known distribution, e.g., Gaussian. Similarly, $T_{\theta}$ is consistently taking $x$ as input throughout Equations in the paper while $x$ and $z$ are taken as input in the algorithm (line 4). For example, in Eq. 5, I can see that $x$ is a prior distribution $y$ is a real distribution and the OT term $c$ is applied in between the Gaussian and the generated samples, which I believe is different from the Algorithm.
>
> **A4.**
> We sincerely appreciate the reviewer for the thoughtful comment regarding the missing clarifications and details in our manuscript. We thoroughly revised the main part and appendix of our manuscript to incorporate these previously missing descriptions.
>
> In our notation in Alg. 1, **$X$ represents the input latent noise, sampled from the source (prior) distribution $\mu$. $Y$ denotes the real data, sampled from the target (data) distribution $\nu$.** The additional Gaussian noise $z\sim \mathcal{N}(0,I)$ in Alg. 1 is the auxiliary variable. We introduced this auxiliary variable $z$ to represent the stochastic transport map $T_{\theta}$ in the OT map models, such as UOTM. This auxiliary noise is provided to the generator $T_{\theta}$ to enhance generative performance from the practical perspective. We included a description of this auxiliary variable $z$ in Sec 3.1 as follows:
>
> > Also, the Gaussian noise $z$ represents the auxiliary variable and is different from the input prior noise $x \sim \mu$. This auxiliary variable $z$ is introduced to represent the stochastic transport map $T_{\theta}$ in the OT map models, such as UOTM.
>
>
> $ $
>
> ---
> **Q3.**
> (page 6, “Effect of Cost in Mode Collapse”) It is not straightforward how the cost function actually helps the mode coverage.
>
> **A5.**
> In Sec 3, we observed that the cost function helps the mode coverage of OT-based adversarial networks in both the Toy dataset (Fig 1) and CIFAR-10 (Fig 4). **We hypothesize that this observation stems from the indirect supervision provided by the cost minimization in the OT problem.** More precisely, this cost minimization induces the generator $T$ to transport the generated image $T(x)$ close to the input noise $x$. **Therefore, when dealing with a multimodel data distribution $\nu$, this cost minimization property specifies which part of the source distribution $\mu$ should correspond to each mode of $\nu$, as visualized in Fig 5.** We believe this specification is helpful in addressing mode coverage. On the contrary, when there is no cost function (WGAN and UOTM w/o cost in Fig 5), each input noise is randomly transported to the support of data distribution.
>
>
> $ $
>
> ---
> **Q4.**
> Why the results in Fig. 1 and Fig. 5 are different?
>
> **A6.**
> Fig 1 visualizes the generated samples of each model for every 6K iterations. In Fig 5, we chose the best iterations for each model (from another run), since WGAN and OTM collapse during long training as we can see in Right of Fig 1.
>
>
> $ $
>
> ---
> **Q5.**
> Performance of UOTM-SD in the toy dataset? and Qualitative comparison results in Cifar10 (UOTM-SD v.s. UOTM)?
>
> **A7.**
> In the initial manuscript, we provided the generated samples of UOTM-SD and UOTM on CIFAR-10 in Appendix D.5. We added additional clarifications on Page 9, indicating these qualitative samples as follows:
>
> > See Appendix D.5 for the qualitative comparison of generated samples.
>
> Furthermore, we included UOTM-SD results on the toy dataset in Appendix D.1. In the Toy dataset, UOTM and UOTM-SD demonstrate almost similar results.

---

> > ### Author Response · Authors · 2023-11-22
> >
> > We thank the reviewer for the efforts in reviewing our paper. We would appreciate it if the reviewer let us know whether our response was helpful in addressing the reviewer's concerns. If there are additional concerns or questions, please let us know.

---

### Meta-Review · Area_Chair_b31G · 2023-12-08

**Metareview:**

The authors propose to consider a unifying framework for OT-based adversarial networks. The authors provide a detailed analysis on components of OT-based adversarial network approaches which guides to a simple but novel proposed approach (based on observations on standard OT and unbalanced OT). The authors illustrate advantages of the proposed method on CIFAR-10. The Reviewers agree that it is an interesting contribution with detailed analysis. However, the authors also raised concerns about theoretical contribution the unified framework which may not offer new insights, but only serve as a guidance for experiments. Additionally, the further empirical results during rebuttal help to improve the experiments for the submission. The authors should incorporate them into the update. Overall, the submission is essentially on the borderline. In case, there is still space, the submission may be a candidate.

**Justification For Why Not Higher Score:**

The strong part of the submission is to give a detailed analysis which gives guidances for experiments. However, some raised issues on theory still remains, the developed theory does not have new insight, or why it is necessary to develop the proposed approach.

**Justification For Why Not Lower Score:**

I think the submission is on the borderline. It may be a candidate for acceptance in case there is still some space.

---

### Decision · Program_Chairs · 2024-01-16

Accept (poster)